# INDEPENDENT COMPONENT ALIGNMENT
# FOR MULTI-TASK LEARNING

## ABSTRACT

We present a novel gradient-based multi-task learning (MTL) approach that balances training in multi-task systems by aligning the independent components of the training objective. In contrast to state-of-the-art MTL approaches, our method is stable and preserves the ratio of highly correlated tasks gradients. The method is scalable, reduces overfitting, and can seamlessly handle multi-task objectives with a large difference in gradient magnitudes. We demonstrate the effectiveness of the proposed approach on a variety of MTL problems including digit classification, multi-label image classification, camera relocalization, and scene understanding. Our approach performs favourably compared to other gradient-based adaptive balancing methods, and its performance is backed up by theoretical analysis.

## 1 INTRODUCTION

In multi-task learning (MTL, *e.g.*, Caruana, 1993; Doersch & Zisserman, 2017), several tasks are optimized simultaneously using a single model and by leveraging shared information across tasks to improve generalization and boost performance for all tasks. MTL is a key component in real-world applications, such as natural language processing (Liu et al., 2019c), computer vision (Kendall et al., 2015; Kokkinos, 2017), and reinforcement learning (Parisotto et al., 2016; Teh et al., 2017). However, these systems are difficult to train since different tasks need to be properly balanced, which is challenging when the multi-task gradient is dominated by the gradient of one of the tasks.

MTL approaches have demonstrated rapid progress, and in general, they can be divided into two categories. The first group of methods aims at designing multi-task neural network architectures which should learn both task-shared and task-specific features by using hard and soft parameter sharing (Liu et al., 2019b; Lu et al.; Houlsby et al., 2019; Pfeiffer et al., 2021). The methods in the second group focus on multi-task optimization strategies. To learn all tasks with equal importance, some methods use gradient replacement (Yu et al., 2020; Lopez-Paz & Ranzato, 2017) or balance the individual loss functions with weights sampled from a uniform distribution, learned by using task-dependant uncertainty (Kendall et al., 2018), determined by utilising the learning speed for the task (Liu et al., 2019a;b; Lee & Son, 2020), or obtained by minimizing an auxiliary loss (Chen et al., 2018). However, combining individual loss functions into one using a weighted average can be hard and prone to errors if some task gradients are in conflict with each other. To alleviate this limitation, a variety of multi-objective optimization methods have been proposed (Désidéri, 2012; Sener & Koltun, 2018). They aim to find Pareto-stationary solutions for which the performance of any of the objectives can only be improved by deteriorating performance of some other objective. Although these methods demonstrate good performance, their main limitation is local convergence. Specifically, if the objective function is non-convex, which is typical in deep learning, the methods converge to a poor Pareto-stationary solution. Moreover, there is a gap between optimization and generalization performance as proven by Lin et al. (2019).

This work hinges on the realization that improving upon the overall performance in multi-task learning can be formulated by conservatism in the individual learning steps of uncorrelated losses. We formulate an approach that aims to guarantee consistent improvement over dynamically constructed linearly independent tasks at every step. This results in a multi-objective optimization algorithm that automatically balances training in multi-task systems by aligning orthogonal components of the objective. We address the problem of a large difference in the task gradient magnitudes through the condition number of the gradient matrix. This number defines the dominance rate in the multi-

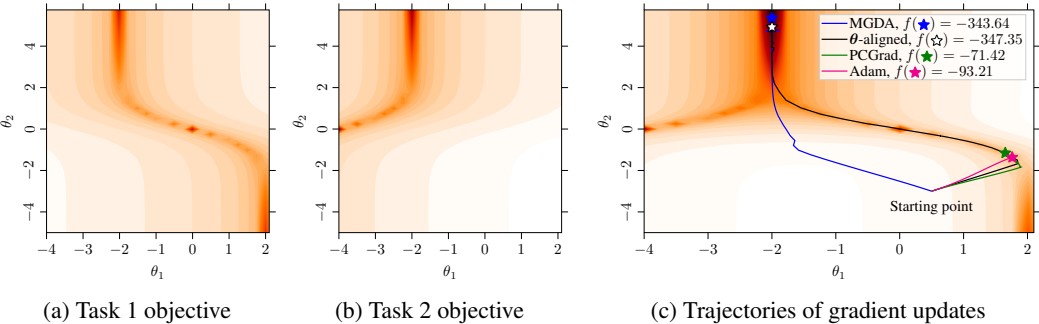

Figure 1: **Synthetic multi-task problem.** Following Yu et al. (2020), we evaluate our approach and recent baseline MTL methods on a multi-task optimization problem, where the multi-task objective gradient is dominated by one task. For each MTL method, we visualize the trajectory of gradient updates computed for an equal number of iterations. The proposed $\theta$-aligned method can handle the imbalance and converges faster compared to prior state-of-the-art MTL approaches.

objective task and is obtained as the ratio of maximum and minimum singular values. Fig. 1 illustrates the proposed approach and shows its performance compared to other gradient-based MTL methods when the difference in the task gradient magnitudes is large.

The contributions of this paper are summarized as follows. *(i)* We present a novel gradient-based MTL approach that dynamically constructs consistent losses and seamlessly handles the problem of a large difference in task gradients. *(ii)* We provide a computationally efficient approximation for the approach with constant time complexity and which is directly applicable to large-scale problems. *(iii)* We formulate a new metric, the dominance rate, as an interpretable way of establishing the maximum possible dominance in the gradient system. Additionally, we perform an extensive evaluation of the proposed approach on various MTL benchmarks and demonstrate that our method outperforms prior state-of-the-art MTL methods.

## 2 RELATED WORK

A multi-task formulation is well motivated and ubiquitous to many application domains such as computer vision (Bilen & Vedaldi, 2016; Kokkinos, 2017; Zamir et al., 2018; Nekrasov et al., 2019; Kendall et al., 2018), natural language processing (Collobert & Weston, 2008; Luong et al., 2015; Dong et al., 2015), speech processing (Seltzer & Droppo, 2013), and robotics (Wulfmeier et al., 2020; Levine et al., 2016). In this work we focus on gradient-based malti-task learning strategies and refer the interested reader to comprehensive surveys by Caruana (1993), Ruder (2017), and Crawshaw (2020) for a more detailed overview. One way to ease multi-task optimization is to balance the individual loss functions for different tasks. Prior methods to simultaneously learning multiple tasks parameterize the total objective as a weighted sum of task-specific loss functions whose weighting coefficients are manually tuned (Kendall et al., 2015; Laskar et al., 2017). However, using grid search for an optimal weighting is computationally inefficient especially when the number of tasks is large. To address this limitation, Kendall et al. (2018) utilize homoscedastic uncertainty of each task to derive a weighted multi-task loss. Chen et al. (2018) propose a heuristic to tune loss weights based on task gradient magnitudes.

One of the main challenges in MTL is the presence of conflicting gradients when tasks have gradients pointing in opposing directions, *i.e.* negative transfer (Lee et al., 2018). Among other approaches to mitigate the problem of negative transfer, the methods based on explicit gradient modulation have shown great potential (Lopez-Paz & Ranzato, 2017; Yu et al., 2020; Chaudhry et al., 2019). The core idea is to replace a task gradient which conflicts with some other task by a modified, non-conflicting, gradient. Yu et al. (2020) propose a 'gradient surgery' technique which makes the new gradient non-conflicting with the average of task gradients at the previous step. However, it requires checking multiple gradients for conflicts at each update step leading to a large computational overhead.

On the other hand, multi-objective optimization (MOO) is the process of optimizing a set of possibly contrasting objectives simultaneously (Fliege & Svaiter, 2000; Schäffler et al., 2002; Désidéri, 2012;

Peitz & Dellnitz, 2018; Poirion et al., 2017). These methods find a direction that decreases all objectives relying on multi-objective Karush–Kuhn–Tucker (KKT) conditions (Kuhn & Tucker, 1951). The work by Sener & Koltun (2018) extends the classical MGDA (Désidéri, 2012) method to a form that scales well to high-dimensional problems. However, it converges to a non-optimal Pareto-stationary solution if the objective function is non-convex. We empirically compare our approach to a number of MOO methods and demonstrate its superior performance.

## 3 PROBLEM FORMULATION AND NOTATION

Multi-task learning is concerned with finding the set of parameters for a parametric model that correspond to a high average performance across all tasks. More formally, we consider a MTL problem defined over the input space $\mathcal{X}$ and a series of targets $\{\mathcal{Y}^1, \dots, \mathcal{Y}^T\}$, where $T$ is the number of tasks. We consider a parametric model $f(\boldsymbol{x}; \boldsymbol{\theta}^{\text{sh}}, \boldsymbol{\theta}^1, \dots, \boldsymbol{\theta}^T)$ with shared, $\boldsymbol{\theta}^{\text{sh}}$, and task-specific parameters, $\boldsymbol{\theta}^i$. Each task has a corresponding differentiable loss function $\mathcal{L}^i(\hat{\boldsymbol{y}}^i, \boldsymbol{y}^i)$ : $\mathcal{Y}^i \times \mathcal{Y}^i \to \mathbb{R}$. Moreover, we can include prior knowledge on user preferences $p(\mathcal{Y})$ in the model. The MTL problem can be formulated as an empirical risk minimisation (ERM) problem on a data set $\{(\boldsymbol{x}_j, \boldsymbol{y}_j^1, \dots, \boldsymbol{y}_j^T)\}_{j=1}^N$ of i.i.d. data points:

$$\min_{\substack{\boldsymbol{\theta}^{\text{sh}} \\ \boldsymbol{\theta}^1, \dots, \boldsymbol{\theta}^T}} \mathbb{E}_{\mathcal{Y}^i \sim p(\mathcal{Y})} \big[ \mathcal{L}^i(\boldsymbol{\theta}^{\text{sh}}, \boldsymbol{\theta}^i) \big] = \min_{\substack{\boldsymbol{\theta}^{\text{sh}} \\ \boldsymbol{\theta}^1, \dots, \boldsymbol{\theta}^T}} \sum_{i=1}^T w^i \mathcal{L}^i(\boldsymbol{\theta}^{\text{sh}}, \boldsymbol{\theta}^i) = \min_{\substack{\boldsymbol{\theta}^{\text{sh}} \\ \boldsymbol{\theta}^1, \dots, \boldsymbol{\theta}^T}} \sum_{i=1}^T w^i \mathcal{L}^i(\boldsymbol{\theta} = \boldsymbol{\theta}^{\text{sh}}). \quad (1)$$

The vector of preferences $\boldsymbol{w}$ defines the importance of each task: $\boldsymbol{w} = \mathbf{1}$ for MTL; $\boldsymbol{w} \neq \mathbf{1}$ for auxiliary task learning. We consider shared parameters, unless otherwise specified. Since the gradients of $\boldsymbol{\theta}^i$ and $\boldsymbol{\theta}^{\text{sh}}$ are linearly independent pairwise, task-specific parameters have a single gradient, and thus can be omitted without loss of generality.

## 4 INDEPENDENT COMPONENT ALIGNMENT

One of the key challenges in MTL is overfitting to one of the tasks caused by the dominance of the gradient of some linear component in the loss function. In practice, the overfitting might be defined by a dominating direction which has the fastest learning rate compared to others. This dominance can be caused by, for example, gradient interference between the individual loss functions or a significant difference in their magnitudes. Such a situation is illustrated in Fig. 2a, where the gradient for task 2 dominates task 1. Recent works on the interactions of individual tasks (Zamir et al., 2018; 2020) demonstrate that tasks may have a common component. This means that the uniqueness of a particular task is determined by another, unique, component independent to the common one. In this regard, the interference of gradients leads to a faster training rate of the joint subtask reducing the efficiency of learning unique components for each task. Therefore, we need to analyze the independent components of the *gradient system*, rather than the individual gradients themselves. We define dominance in the system of gradients as follows:

**Definition 1** (Dominance rate). ***The rate of dominance*** *of the system of loss functions is the condition number of the matrix composed of the gradients of these functions over the shared parameters:*

$$\mathcal{D}(\mathcal{L}^1(\boldsymbol{\theta}), \dots, \mathcal{L}^T(\boldsymbol{\theta})) = \mathcal{D}(\boldsymbol{G}_{\boldsymbol{\theta}}) = \frac{\sigma_{\max}(\boldsymbol{G}_{\boldsymbol{\theta}})}{\sigma_{\min}(\boldsymbol{G}_{\boldsymbol{\theta}})}, \quad where \quad \boldsymbol{G}_{\boldsymbol{\theta}} = \big(\nabla_{\boldsymbol{\theta}} \mathcal{L}^1(\boldsymbol{\theta}), \dots, \nabla_{\boldsymbol{\theta}} \mathcal{L}^T(\boldsymbol{\theta})\big). \quad (2)$$

This definition is based on the singular value decomposition (SVD) and has a geometric interpretation. The dominance coefficient defines the maximum possible dominance in the gradient system and is obtained as the ratio of the maximum and minimum learning rates of the orthonormal components of the gradient system (as in Fig. 2a). Recently, Wang et al. (2020) propose a metric, OGR to quantitatively estimate the significance of overfitting of multi-modal networks for classification. Although related, the metric is obtained by using an assumption that target distribution is well approximated by a validation data set. In contrast, our measure does not require such a strong prior and solely based on gradients with respect to the training set.

### 4.1 MODEL-AGNOSTIC ALIGNMENT

For effective training over all tasks, the system of gradients of loss functions should be consistent and have a small dominance rate. We take this as a design principle, and aim for avoiding domination and

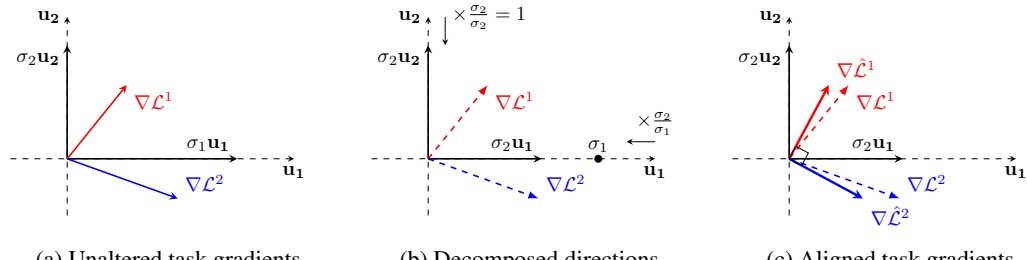

(a) Unaltered task gradients     (b) Decomposed directions     (c) Aligned task gradients

Figure 2: **Geometric interpretation.** (a) Decomposition of the original problem to orthogonal components through a SVD. (b) Re-weighting the gradient directions by balancing the conditioning. (c) The aligned task gradients defining the step directions. Here, $\boldsymbol{u}_i, \sigma_i$ are left singular vector and singular value of gradient matrix $\boldsymbol{G}_{\boldsymbol{\theta}}$, respectively.

inconsistency altogether. We propose constructing a new (consistent) gradient system that best (under conditions that follow) describes the given directions, but whose dominance coefficient is equal to minimum dominance over the task gradients, *i.e.* unit dominance. More formally, we define a new aligned gradient as follows.

**Definition 2** ($\boldsymbol{\theta}$-aligned)**.** *We define a $\boldsymbol{\theta}$-aligned gradient as $\nabla_{\boldsymbol{\theta}}^{\boldsymbol{\theta}a}\mathcal{L}(\boldsymbol{\theta}) \coloneqq \sigma_{\min}(\boldsymbol{G}_{\boldsymbol{\theta}})\hat{\boldsymbol{G}}_{\boldsymbol{\theta}}\boldsymbol{w}$, where vector $\boldsymbol{w}$ denotes a pre-selected task preference and $\hat{\boldsymbol{G}}_{\boldsymbol{\theta}}$ is the solution to the* Procrustes *problem:*

$$\hat{\boldsymbol{G}}_{\boldsymbol{\theta}} = \arg\min_{\boldsymbol{Q}} \|\boldsymbol{G}_{\boldsymbol{\theta}} - \boldsymbol{Q}\|_F, \quad s.t. \quad \boldsymbol{Q}^\top\boldsymbol{Q} = \boldsymbol{I}. \tag{3}$$

In order to calculate the $\boldsymbol{\theta}$-aligned gradient, the solution to the general Procrustes problem is needed. Schönemann (1966) shows that the problem has an analytical solution, namely:

$$\hat{\boldsymbol{G}}_{\boldsymbol{\theta}} = \boldsymbol{U}\boldsymbol{V}^\top, \quad \text{where} \quad \boldsymbol{U}, \boldsymbol{V} : \boldsymbol{G}_{\boldsymbol{\theta}} = \boldsymbol{U}\boldsymbol{\Sigma}\boldsymbol{V}^\top \text{(given by a SVD)}. \tag{4}$$

The analytical solution allows us to avoid the costly procedure for finding the minimum, which reduces the computational cost of our approach. Additionally, this leads to the dominance rate of the resulting matrix being equal to unit dominance: $\mathcal{D}(\sigma_{\min}(\boldsymbol{G}_{\boldsymbol{\theta}})\hat{\boldsymbol{G}}_{\boldsymbol{\theta}}) = \frac{\sigma_{\min}}{\sigma_{\min}} = 1$, *i.e.* we have eliminated the possibility of uneven learning of different tasks. Fig. 2c shows a sketch of the geometric intuition of the aligned gradients after the alignment step.

At this stage we aim to verify that standard gradient descent with $\boldsymbol{\theta}$-aligned gradients corresponds to a sensible optimization procedure. We provide the following analysis, which states the convergence for general non-convex loss functions with standard assumptions on regularity. For a proof and more theoretical results, see App. B.

**Theorem 1.** *Assume $\mathcal{L}^1(\boldsymbol{\theta}), \ldots, \mathcal{L}^T(\boldsymbol{\theta})$ are lower bounded continuously differentiable functions with Lipschitz continuous gradients with $\Lambda > 0$. Assume that the total loss function $\mathcal{L}(\boldsymbol{\theta})$ is a weighted sum of these functions with a pre-selected vector of preferences $\boldsymbol{w}$. Gradient descent with step size $\alpha \leq \frac{1}{\Lambda}$ and $\boldsymbol{\theta}$-aligned gradient converges if the initial rate of dominance is finite, $\mathcal{D}(\boldsymbol{G}_{\boldsymbol{\theta}}) < C$, and task preferences vector $\boldsymbol{w}$ is consistent with the gradient system, $\|\boldsymbol{V}\boldsymbol{w}\| \geq \varepsilon$.*

### 4.2 EFFICIENT ALIGNMENT FOR ENCODER–DECODER NETWORKS

The main limitation of the $\boldsymbol{\theta}$-aligned approach is the requirement of multiple backward passes through the shared part of the model to calculate the gradient matrix. The backward passes are computationally demanding, which results in linear scaling of the training time with respect to the number of tasks. Therefore, it is challenging to apply the method to large-scale learning problems when the number of tasks is high. To address this limitation, we propose an efficient approximation to the $\boldsymbol{\theta}$-aligned approach, obtained under stronger assumptions on the model architecture.

Suppose that the class of hypotheses consists of factorizable functions, *i.e.*

$$f(\boldsymbol{x}; \boldsymbol{\theta}^{\mathrm{sh}}, \boldsymbol{\theta}^1, \ldots, \boldsymbol{\theta}^T) = \big(f(g(\boldsymbol{x}, \boldsymbol{\theta}^{\mathrm{sh}}), \boldsymbol{\theta}^1), \ldots, f(g(\boldsymbol{x}, \boldsymbol{\theta}^{\mathrm{sh}}), \boldsymbol{\theta}^T)\big) \tag{5}$$

where $g(\cdot, \boldsymbol{\theta}^{\text{sh}})$ is a representation function common to all tasks and $f(\cdot, \boldsymbol{\theta}^i)$ is a function that implements model functions for a specific task. This assumption corresponds to encoder–decoder type neural network architectures which are widely used in computer vision applications (*e.g.*, Kendall et al., 2018; Sener & Koltun, 2018; Zamir et al., 2018).

We consider the original Procrustes problem (Def. 2) with new assumptions. We denote the encoded representation as $\boldsymbol{Z} = g(\boldsymbol{x}, \theta^{\text{sh}}) \coloneqq g(\boldsymbol{x}, \theta)$. Suppose the Jacobian $\boldsymbol{J} = \frac{\partial \boldsymbol{Z}}{\partial \boldsymbol{\theta}}$ affects equally all task gradients, *i.e.* $\boldsymbol{J}^\top \boldsymbol{J} = \lambda \boldsymbol{I}$. This assumption is strong, but it is supported by empirical evidence (as also sketched out in Fig. 3) and holds with some accuracy for a wide class of networks.

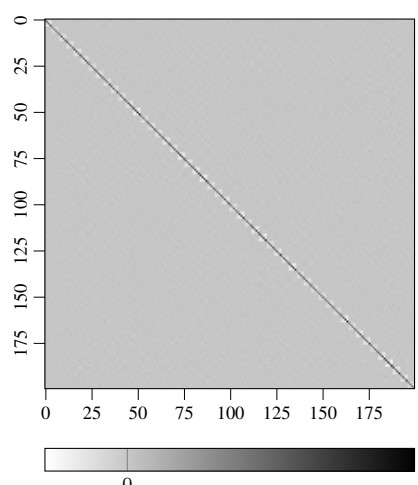

Figure 3: Sketch of $\boldsymbol{J}^\top \boldsymbol{J} \approx \lambda \boldsymbol{I}$. The Jacobian matrix $\boldsymbol{J}$ was computed for the first 50 channels from the output of layer 4 of ResNet-18.

Following these assumptions, the original problem is formulated as follows:

$$\|\boldsymbol{G_\theta} - \boldsymbol{Q}\|_F^2 = \|\boldsymbol{J}^\top \boldsymbol{G_\theta} - \boldsymbol{J}^\top \boldsymbol{Q}\|_F^2 = \lambda^2 \|\boldsymbol{G_Z} - \boldsymbol{P}\|_F^2,$$
$$\text{s.t.} \quad \boldsymbol{P}^\top \boldsymbol{P} = \boldsymbol{I}. \quad (6)$$

In practice, computing the matrix $\boldsymbol{G_Z}$ requires only gradients over the latent representation, which can be computed in one backward pass. Therefore, we define an efficient method of gradient alignment for encoder–decoder networks as follows.

**Definition 3** ($\boldsymbol{Z}$-aligned). *We define a $\boldsymbol{Z}$-aligned gradient as* $\nabla_{\boldsymbol{\theta}}^{\boldsymbol{Z}a} \mathcal{L}(\boldsymbol{\theta}) \coloneqq \sigma_{\min}(\boldsymbol{G_Z}) \frac{\partial \boldsymbol{Z}}{\partial \boldsymbol{\theta}} \hat{\boldsymbol{G}}_{\boldsymbol{Z}} \boldsymbol{w}$, *where $\boldsymbol{w}$ denotes a pre-selected task preference and $\hat{\boldsymbol{G}}_{\boldsymbol{Z}}$ is the solution to the Procrustes problem:*

$$\hat{\boldsymbol{G}}_{\boldsymbol{Z}} = \arg\min_{\boldsymbol{P}} \|\boldsymbol{G_Z} - \boldsymbol{P}\|_F, \quad \text{s.t.} \quad \boldsymbol{P}^\top \boldsymbol{P} = \boldsymbol{I}. \quad (7)$$

The value of the rate of dominance in this case will strongly depend on the condition for the Jacobian. The value of the growth rate $\lambda$ does not affect the value of the dominance coefficient:

$$\mathcal{D}(\sigma_{\min}(\boldsymbol{G_Z}) \boldsymbol{J} \hat{\boldsymbol{G}}_{\boldsymbol{Z}}) \approx \frac{\lambda \sigma_{\min}(\boldsymbol{G_Z})}{\lambda \sigma_{\min}(\boldsymbol{G_Z})} = 1. \quad (8)$$

---

**Algorithm 1** Gradient descent with alignment both for the $\boldsymbol{\theta}$-aligned and $\boldsymbol{Z}$-aligned variants.

---
1: **for** $i = 1$ **to** $T$ **do**
2:     $\boldsymbol{\theta}_{t+1}^i = \boldsymbol{\theta}_t^i - \eta \nabla_{\boldsymbol{\theta}^i} \hat{\mathcal{L}}^i(\boldsymbol{\theta}^{\text{sh}}, \boldsymbol{\theta}^i)$         $\triangleright$ Gradient descent on task-specific parameters
3: **end for**
4: **if** $\boldsymbol{\theta}$-alignment **then**
5:     Compute $\boldsymbol{G_\theta} = \left( \nabla_{\boldsymbol{\theta}^{\text{sh}}} \mathcal{L}^1(\boldsymbol{\theta}), \dots, \nabla_{\boldsymbol{\theta}^{\text{sh}}} \mathcal{L}^T(\boldsymbol{\theta}) \right)$
6:     $\hat{\boldsymbol{G}}_{\boldsymbol{\theta}}, \sigma = \textsc{ProcrustesSolver}(\boldsymbol{G_\theta})$         $\triangleright$ Solve problem in Def. 2
7:     $\nabla_{\boldsymbol{\theta}} \mathcal{L}(\boldsymbol{\theta}) = \sigma \hat{\boldsymbol{G}}_{\boldsymbol{\theta}} \boldsymbol{w}$         $\triangleright$ Compute $\boldsymbol{\theta}$-aligned gradient
8: **else if** $\boldsymbol{Z}$-alignment **then**
9:     Compute $\boldsymbol{G_Z} = \left( \nabla_{\boldsymbol{Z}} \mathcal{L}^1(\boldsymbol{\theta}), \dots, \nabla_{\boldsymbol{Z}} \mathcal{L}^T(\boldsymbol{\theta}) \right)$
10:    $\hat{\boldsymbol{G}}_{\boldsymbol{Z}}, \sigma = \textsc{ProcrustesSolver}(\boldsymbol{G_Z})$         $\triangleright$ Solve problem in Def. 3
11:    $\nabla_{\boldsymbol{\theta}} \mathcal{L}(\boldsymbol{\theta}) = \sigma \boldsymbol{J} \hat{\boldsymbol{G}}_{\boldsymbol{Z}} \boldsymbol{w}$         $\triangleright$ Compute $\boldsymbol{Z}$-aligned gradient
12: **end if**
13: $\boldsymbol{\theta}_{t+1}^{\text{sh}} = \boldsymbol{\theta}_t^{\text{sh}} - \eta \nabla_{\boldsymbol{\theta}} \mathcal{L}(\boldsymbol{\theta})$         $\triangleright$ Gradient descent on the shared parameters

14: **procedure** $\textsc{ProcrustesSolver}(\boldsymbol{G})$
15:     Factorize $\boldsymbol{G}$, $\boldsymbol{G} = \boldsymbol{U} \boldsymbol{\Sigma} \boldsymbol{V}^\top$         $\triangleright$ Compute SVD decomposition
16:     $\hat{\boldsymbol{G}} = \boldsymbol{U} \boldsymbol{V}^\top$         $\triangleright$ Analytical solution of Procrustes problem
17:     Compute $\sigma = \min_i \boldsymbol{\Sigma}_{ii}$         $\triangleright$ Find a minimal learning rate
18:     **return** $\hat{\boldsymbol{G}}, \sigma$
19: **end procedure**

---

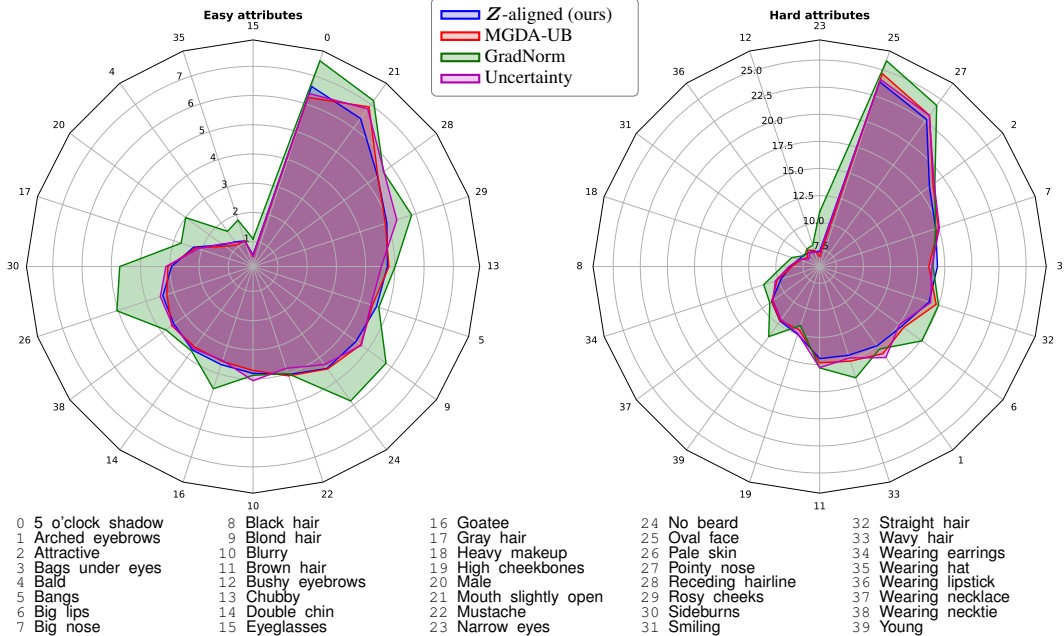

| 0 | 5 o'clock shadow | 8 | Black hair | 16 | Goatee | 24 | No beard | 32 | Straight hair |
|---|---|---|---|---|---|---|---|---|---|
| 1 | Arched eyebrows | 9 | Blond hair | 17 | Gray hair | 25 | Oval face | 33 | Wavy hair |
| 2 | Attractive | 10 | Blurry | 18 | Heavy makeup | 26 | Pale skin | 34 | Wearing earrings |
| 3 | Bags under eyes | 11 | Brown hair | 19 | High cheekbones | 27 | Pointy nose | 35 | Wearing hat |
| 4 | Bald | 12 | Bushy eyebrows | 20 | Male | 28 | Receding hairline | 36 | Wearing lipstick |
| 5 | Bangs | 13 | Chubby | 21 | Mouth slightly open | 29 | Rosy cheeks | 37 | Wearing necklace |
| 6 | Big lips | 14 | Double chin | 22 | Mustache | 30 | Sideburns | 38 | Wearing necktie |
| 7 | Big nose | 15 | Eyeglasses | 23 | Narrow eyes | 31 | Smiling | 39 | Young |

Figure 4: **Performance of MTL methods on CELEBA**. The percentage error per attribute on the CELEBA data set presented as a radar chart, where tighter is better. Following Sener & Koltun (2018), we divide the attributes into two categories: easy (left) and hard (right). See Sec. 5.2 for more details. We present here the methods with comparable time complexity for clear comparison.

Even though our observations confirm that the Jacobian often satisfies our assumptions (see also Sec. 5), we prove (proof of the theorem below in App. B) that gradient descent with $\boldsymbol{Z}$-aligned gradients will converge in a milder setup even if our requirements are not met. Our formulation increases the stability of the proposed approach because in worst case it behaves similarly to gradient descent without alignment.

**Theorem 2.** *Assume $\mathcal{L}^1(\boldsymbol{\theta}), \ldots, \mathcal{L}^T(\boldsymbol{\theta})$ are lower bounded continuously differentiable functions with Lipschitz continuous gradients with $\Lambda > 0$. Assume that the total loss function $\mathcal{L}(\boldsymbol{\theta})$ is a weighted sum of these functions with a pre-selected vector of preferences $\boldsymbol{w}$. The gradient descent with step size $\alpha \leq \frac{1}{\Lambda}$ and a $\boldsymbol{Z}$-aligned gradient converges if the Jacobian $\boldsymbol{J} = \frac{\partial \boldsymbol{Z}}{\partial \theta}$ is full rank, the initial rate of dominance is finite, i.e. $\mathcal{D}(\boldsymbol{G_Z}) < C_\sigma$ and $\mathcal{D}(\boldsymbol{J}) < C_\lambda$, and task preferences vector $\boldsymbol{w}$ is consistent with the gradient system, $\|\boldsymbol{V}\boldsymbol{w}\| \geq \varepsilon$.*

Alg. 1 summarizes the approaches for the $\boldsymbol{\theta}$-aligned and the $\boldsymbol{Z}$-aligned multi-task learning methods.

## 5 EXPERIMENTS

We empirically demonstrate the effectiveness of the proposed approach on a number of different multi-task learning benchmarks. Specifically, we tackle the problems of scene understanding, multi-digit, and multi-label classification by evaluating our method on the CITYSCAPES (Cordts et al., 2016), MULTIMNIST (Sabour et al., 2017), and CELEBA (Liu et al., 2015) data sets, respectively. Furthermore, we experiment with image-based localization by jointly predicting camera orientation and translation on the 7SCENES data set (Shotton et al., 2013).

**Baselines.** In this work, we consider the following MTL baseline methods: *(1)* **uniform scaling**: optimizing a uniformly weighted sum of individual task objectives, *i.e.* $\frac{1}{T} \sum_t \mathcal{L}^t$; *(2)* **Uncertainty** (Kendall et al., 2018): using uncertainty to find weighting coefficients for each task; *(3)* **MGDA** (Désidéri, 2012): using the gradients of each task to solve an optimization problem for an update; *(4)* **MGDA-UB** (Sener & Koltun, 2018): optimizing an upper bound for the MGDA optimization objective; *(5)* **GradNorm** (Chen et al., 2018): normalizing the gradients to balance the learning of multiple tasks, and *(6)* **PCGrad** (Yu et al., 2020): performing gradient projection to avoid

Table 2: MULTIMNIST multi-task results. For each task, we highlight the best performing method in **bold**. We also report the coefficient of dominance for each MTL approach.

| Method | Dominance rate $\mathcal{D}$ (Def. 1) | | | | Accuracy | |
| | Mean | Std | Max | Min | Task-L [%] | Task-R [%] |
|---|---|---|---|---|---|---|
| MGDA | 1.63 | 1.45 | 38.03 | 1.004 | 96.83 | 95.06 |
| MGDA-UB | 1.41 | 0.36 | 4.00 | 1.004 | 96.76 | 94.96 |
| PCGrad | 1.45 | 0.37 | 3.76 | 1.005 | 97.04 | 95.29 |
| GradNorm | 3.12 | 0.83 | 8.91 | 1.704 | 96.51 | 95.31 |
| Uncertainty | 1.47 | 0.44 | 5.17 | 1.007 | 97.03 | 95.06 |
| $\boldsymbol{\theta}$-aligned (ours) | 1.00 | 0.0 | 1.00 | 1.000 | **97.10** | **95.45** |
| $\boldsymbol{Z}$-aligned (ours) | 1.36 | 0.29 | 3.17 | 1.005 | 96.98 | **95.45** |

the negative interactions between tasks gradients. The proposed approach and the baseline methods are implemented using the PyTorch framework (Paszke et al., 2019).

**Analysis of computational complexity.** The main computational cost of all algorithms depends on the number of backward passes over the shared parameters for each task and for each iteration of the gradient descent. We denote the number of tasks as $T$ and the number of iterations in the Frank–Wolfe solver (Jaggi, 2013; Sener & Koltun, 2018) as $K$. By Def. 2, the $\boldsymbol{\theta}$-aligned gradient requires only the precomputed matrix $\boldsymbol{G_\theta}$ and therefore, it performs $T$ backward passes on one step. In contrast, the $\boldsymbol{Z}$-aligned gradient can be computed in a single pass, which is the best possible case. The computational cost for each MTL method is provided in Table 1. We consider the time complexity of the SVD decomposition constant.

Table 1: Computational cost per step for $T$ tasks and $K$ steps.

| Method | Cost |
|---|---|
| MGDA | $\mathcal{O}(T + K)$ |
| MGDA-UB | $\mathcal{O}(K)$ |
| PCGrad | $\mathcal{O}(T)$ |
| GradNorm | $\mathcal{O}(T)$ |
| Uncertainty | $\mathcal{O}(1)$ |
| $\boldsymbol{\theta}$-aligned (ours) | $\mathcal{O}(T)$ |
| $\boldsymbol{Z}$-aligned (ours) | $\mathcal{O}(1)$ |

## 5.1 SYNTHETIC MULTI-TASK PROBLEM

To illustrate the proposed method on a simple problem, we consider a multi-task optimization objective consisting of two tasks presented in Fig. 1 which was originally proposed by Yu et al. (2020). Specifically, the multi-task gradient of the objective is dominated by the gradient of one of the tasks leading to suboptimal training dynamics. As a result, the standard optimizers (*e.g.*, Adam) improve only the dominating task (the magenta trajectory in Fig. 1c) and struggle to optimize the whole objective. In contrast, the proposed approach is more stable and outperforms MGDA (Désidéri, 2012) and PCGrad (Yu et al., 2020). Moreover, it demonstrates favourable convergence rate compared to the baselines: 5k and 500k iterations are required for our method and PCGrad, respectively.

## 5.2 MULTI-TASK CLASSIFICATION

We tackle the problem of multi-task classification and evaluate the proposed method on MULTIMNIST digit classification and celebrity face image data in CELEBA.

**MULTIMNIST.** Following the same procedure proposed by Sener & Koltun (2018), we convert digit classification into a multi-digit classification problem. Specifically, for each image of the original MNIST data set, we randomly sample a different image from a uniform distribution. The two images are then overlaid in such a way that one of the images is put at the top-left and the other one at the bottom-right. Similarly to Sener & Koltun (2018); Yu et al. (2020), we generate 60k such images and directly evaluate the proposed approach and the baselines. The resulting task in the multi-task classification problem is to classify the digit on the top left (task-L) and bottom right (task-R), respectively. The results are summarized in Table 2. The proposed MTL approach $\boldsymbol{\theta}$-aligned surpasses all baselines by a noticeable margin: up to **+0.06%** for Task-L and **+0.16%** for Task-R compared to the best performing contender. To emphasize the significance of the gradient dominance problem and its influence on the results, we report the coefficient of dominance (Def. 1) for each method, which shows large variability for the baselines.

Table 3: Scene understanding and multi-label classification performance. The best score is in **bold** and the second best score is in *italic*. For CELEBA, we report the mean±std over 10 random seeds.

| | Scene understanding (CITYSCAPES) | | | CELEBA |
|---|---|---|---|---|
| | Segmentation mIoU [%] ↑ | Instance error L1 [px] ↓ | Disparity error MSE (up to scale) ↓ | mAcc [%] ↑ |
| MGDA | *66.72* | 17.02 | 0.3294 | $90.49 \pm 0.05$ |
| MGDA-UB | 66.37 | 18.63 | **0.3212** | $90.91 \pm 0.02$ |
| GradNorm | 57.24 | 10.29 | 0.3536 | $91.18 \pm 0.04$ |
| PCGrad | 54.06 | *9.91* | 0.3837 | $\mathbf{91.53 \pm 0.05}$ |
| Uncertainty | 60.12 | **9.87** | 0.3314 | $91.33 \pm 0.03$ |
| $\theta$-aligned (ours) | **66.98** | 10.81 | 0.3266 | $91.32 \pm 0.04$ |
| $Z$-aligned (ours) | 66.07 | 10.54 | *0.3213* | $91.36 \pm 0.05$ |

**CELEBA.** We try our method on a multi-label classification problem by using the CELEBA data set (Liu et al., 2015), which consists of 200k face images annotated with 40 attributes. Each attribute can be obtained by solving a binary classification problem and thus we convert it to 40-way MTL following (Sener & Koltun, 2018; Yu et al., 2020). For all of the methods we use the same CNN architecture based on ResNet-18 as proposed by Sener & Koltun (2018). The binary classification error averaged over all 40 attributes is used for evaluating performance and is reported in Table 3 (mean $\pm$ std over 10 random seeds). Our approach achieves **91.36%** classification accuracy, which outperforms other strong methods such as MGDA-UB (Sener & Koltun, 2018) and Uncertainty (Kendall et al., 2018), indicating the effectiveness of the proposed method when the number of tasks is large. The percentage of error per attribute is illustrated in Fig. 4.

## 5.3 SCENE UNDERSTANDING

We also apply the proposed MTL approach to scene understanding. Specifically, we aim to jointly solve the problem of semantic segmentation, instance segmentation, and depth estimation. We follow Sener & Koltun (2018) and use the CITYSCAPES data sets (Cordts et al., 2016) consisting of video frames captured in the streets of urban cities and labelled with instance, semantic segmentations, and depth maps. Similarly to Kendall et al. (2018), we use a CNN model with a UNet architecture comprising a shared encoder based on ResNet-50 and three independent task-specific decoders. In our experiments, we resize all training and validation images to $256 \times 512$ and use standard pixel-wise loss functions for each task: negative log likelihood loss for semantic segmentation and L1 loss for instance segmentation and depth estimation. In Table 3, we benchmark our approach in all three tasks and show that the proposed method performs favourably compared to other state-of-the-art MTL methods.

## 5.4 CAMERA RELOCALIZATION

Finally, we evaluate the performance of the proposed MTL approach on the task of image-based localization. Image-based localization, or camera relocalization refers to the problem of estimating the 6 degree-of-freedom (DoF) camera pose from visual data with respect to a known environment. It has been shown that camera relocalization can be solved by casting it as a regression problem (Kendall et al., 2018; Laskar et al., 2017; Melekhov et al., 2017; Walch et al., 2017; Noha et al., 2018), where a 7-dimensional camera pose vector $p = [t, r]$ is directly estimated by a CNN. The camera pose vector $p$ consists of a translation component $t = [t_1, t_2, t_3]$ and an orientation component $r = [q_1, q_2, q_3, q_4]$, represented by a quaternion. Following Kendall et al. (2018), we utilize the Microsoft 7SCENES (Shotton et al., 2013) data set which is a common benchmark for indoor pose regression methods. The data set comprises RGB-D image sequences of seven different indoor scenes from a handheld Kinect camera, and each scene provides training and testing image sequences consisting of 1000–7000 frames at $640 \times 480$ resolution. Similar to Walch et al. (2017); Laskar et al. (2017), we use ResNet-34 as a backbone network in all our experiments to make the comparison between the MTL methods fair. For each method, we compute the median error of camera translation, $t$, and orientation, $r$, per scene. We also report the growth rate, $\mathcal{R}$, as a measure of improvement of multi-task learning methods over the uniform scaling approach. It is defined as $\mathcal{R} = 1 - \frac{P_i}{P}$ where $P_i$ is the mean translation (orientation) error of a MTL method; $P$ is the mean translation (orientation)

Table 4: Camera relocalization performance of the proposed method and existing MTL approaches for the 7SCENES data set. We report median translation $t$, orientation $r$ errors and the growth rate, $\mathcal{R}$ for each scene. For each scene, we highlight the best performing method in terms of translation and orientation error in *italic* and **bold**, respectively.

| | | | | | Scenes | | | | | |
|---|---|---|---|---|---|---|---|---|---|---|
| | | CHESS | FIRE | HEADS | OFFICE | PUMPKIN | KITCHEN | STAIRS | Mean | $\mathcal{R}$, % |
| uniform | $t$, m | 0.48 | 1.78 | 0.46 | 0.70 | 0.72 | 0.90 | 0.47 | 0.79 | – |
| | $r$, ° | 5.83 | 11.57 | 13.04 | **8.43** | 6.79 | 8.88 | **11.22** | 9.39 | – |
| MGDA | $t$, m | 0.29 | 0.73 | 0.81 | 0.33 | 0.47 | 0.33 | 0.46 | 0.49 | +37.98 |
| | $r$, ° | 5.90 | 11.84 | 12.59 | 8.67 | **6.37** | 9.08 | 11.46 | 9.42 | −0.32 |
| MGDA-UB | $t$, m | 0.30 | 0.73 | 0.66 | 0.59 | 0.46 | 0.57 | 0.55 | 0.55 | +30.38 |
| | $r$, ° | 5.92 | 11.74 | 12.42 | 9.09 | 6.30 | **8.69** | 11.76 | 9.42 | −0.32 |
| GradNorm | $t$, m | *0.17* | 0.33 | 0.25 | 0.24 | *0.25* | 0.26 | 0.37 | 0.27 | +65.82 |
| | $r$, ° | 7.08 | 13.54 | 14.70 | 9.65 | 8.35 | 9.08 | 14.07 | 10.92 | −16.29 |
| PCGrad | $t$, m | *0.17* | *0.30* | *0.23* | *0.23* | 0.26 | *0.25* | *0.37* | *0.26* | +67.09 |
| | $r$, ° | 9.10 | 13.17 | 16.19 | 10.57 | 9.00 | 10.35 | 13.71 | 11.73 | −24.92 |
| Uncertainty | $t$, m | 0.19 | 0.33 | 0.30 | 0.25 | 0.25 | 0.28 | 0.48 | 0.30 | +62.03 |
| | $r$, ° | 9.32 | 16.15 | 17.86 | 10.83 | 11.22 | 10.20 | 14.60 | 12.88 | −37.17 |
| $\boldsymbol{\theta}$-aligned | $t$, m | 0.18 | 0.46 | 0.69 | 0.27 | 0.34 | 0.29 | 0.61 | 0.41 | +48.10 |
| | $r$, ° | 5.82 | 12.07 | 18.15 | 9.38 | 8.99 | 10.27 | 12.84 | 11.07 | −17.89 |
| $\boldsymbol{Z}$-aligned | $t$, m | 0.18 | 0.48 | 0.43 | 0.28 | 0.27 | 0.32 | 0.42 | 0.34 | +56.96 |
| | $r$, ° | **5.40** | **11.18** | **12.14** | 9.57 | **6.37** | 8.91 | 11.54 | **9.30** | +0.96 |

error of the uniform scaling method. The results are provided in Table 4. In terms of camera orientation error, the proposed $Z$-aligned approach is superior to other MTL methods for the majority of scenes. Regarding camera translation accuracy, $Z$-aligned can outperform MGDA-UB (Sener & Koltun, 2018) and improve the averaged translation error by **0.21** m showing comparable performance to PCGrad (Yu et al., 2020). See App. D.2 for details on the evaluation pipeline and additional results.

## 6 DISCUSSIONS AND CONCLUSIONS

We have presented a new approach to gradient-based multi-task learning (MTL) that balances training in multi-task systems by aligning the independent components of the gradient of the training objective. In Sec. 4.1, we derived the '$\theta$-aligned' approach and the conditions for it to converge (details in App. B). To ensure practicality, we extended our approach to incorporate heuristics motivated by common encode–decoder networks, which was formulated into the '$Z$-aligned' approach in Sec. 4.2. Additionally, we formulate a new metric, the dominance rate (Def. 1) that provides an interpretable way of establishing the maximum dominance in a gradient system and gives insights to pre-existing MTL approaches.

We also performed an extensive set of experiments for analyzing the *practical* benefits of our approach. To gain insights, we provided results for a toy benchmark problem and compared our approach to other recent gradient-based MTL approaches on MULTIMNIST to study the effect of the rate of dominance. As for more pure benchmarking, we provided results for MTL problems in image classification (CELEBA), scene understanding (CITYSCAPES), and camera relocalization (7SCENES), each being large-scale and well-known benchmarks in their respective domains. We show state-of-the-art performance on MULTIMNIST, CITYSCAPES, and 7SCENES, while the differences between methods remains small for CELEBA. The results are particularly interesting given the smaller computational requirements in the $Z$-aligned method (see Table 1).

A reference implementation of the methods presented in this paper is currently available as supplementary material to this submission.

## REPRODUCIBILITY STATEMENT

This paper proposes a new method for multi-task learning, and the contributions of the paper are technical of nature. To back up the statements in the main paper, we have provided proofs for the given theorems in the appendix (included after the references). Additionally, we have also provided further theorems and proofs that help the reader assess the applicability of the proposes methodology to other optimization setups.

We also include a code package as a supplementary zip file to this double-blind submission. This reference implementation helps understand technical details, and the included scripts can be used for reproducing the numerical experiments in the paper. Upon acceptance, the codes will be available as a GitHub repository under the MIT license.

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

APPENDIX

## A  NOTATION

$$
\begin{array}{rl}
\mathcal{L}^i(\boldsymbol{\theta}) & i^{\text{th}} \text{ loss function} \\
\boldsymbol{\theta} & \text{set of a shared parameters} \\
\boldsymbol{Z} & \text{shared representation} \\
\nabla_{\boldsymbol{\theta}}\mathcal{L}^i(\boldsymbol{\theta}) = \nabla\mathcal{L}^i(\boldsymbol{\theta}) & \text{the gradient of loss function over the shared parameters} \\
\nabla_{\boldsymbol{Z}}\mathcal{L}^i(\boldsymbol{\theta}) & \text{the gradient of loss function over the shared representation} \\
\boldsymbol{G}_{\boldsymbol{\theta}} = (\nabla_{\boldsymbol{\theta}}\mathcal{L}^1(\boldsymbol{\theta}), \cdots, \nabla_{\boldsymbol{\theta}}\mathcal{L}^T(\boldsymbol{\theta})) & \text{matrix of gradients over the shared parameters} \\
\boldsymbol{G}_{\boldsymbol{Z}} = (\nabla_{\boldsymbol{Z}}\mathcal{L}^1(\boldsymbol{\theta}), \cdots, \nabla_{\boldsymbol{Z}}\mathcal{L}^T(\boldsymbol{\theta})) & \text{matrix of gradients over the shared representation} \\
\nabla_{\boldsymbol{\theta}}^{\boldsymbol{\theta} a}\mathcal{L}(\boldsymbol{\theta}) & \boldsymbol{\theta}\text{-aligned gradient} \\
\nabla_{\boldsymbol{\theta}}^{\boldsymbol{Z} a}\mathcal{L}(\boldsymbol{\theta}) & \boldsymbol{Z}\text{-aligned gradient} \\
\nabla_{\boldsymbol{\theta}}^{*a}\mathcal{L}(\boldsymbol{\theta}) & \text{common notation for both aligned gradients} \\
\boldsymbol{J} = \frac{\partial \boldsymbol{Z}}{\partial \theta} & \text{Jacobian of shared representation over shared parameters} \\
\boldsymbol{w} & \text{a manual weight vector for cumulative loss function} \\
\langle \cdot, \cdot \rangle & \text{a dot product of vectors} \\
\mathcal{H}_{\boldsymbol{V}} = \operatorname{span}(\boldsymbol{V}) & \text{linear hull spanned on the column vectors of } \boldsymbol{V} \\
\| \cdot \|_F & \text{Frobenius norm}
\end{array}
$$

## B  PROOFS OF THEOREMS

**Synopsis.**  In this theorem we prove that the worst case performance of the $\mathbf{Z}$-aligned gradient is no worse than in standard gradient descent. The constraints mentioned in the theorem always hold in practice.

**Lemma 1.**  *Assume $\mathcal{L}(\boldsymbol{\theta})$ to be continuously differentiable and $\nabla\mathcal{L}(\boldsymbol{\theta})$ to be Lipschitz continuous with $\Lambda > 0$. Then, the following restriction is valid for any descent with the step $\alpha\boldsymbol{r}$:*

$$
\mathcal{L}(\boldsymbol{\theta}_t) - \mathcal{L}(\boldsymbol{\theta}_{t+1}) \geq \alpha\langle \nabla\mathcal{L}(\boldsymbol{\theta}_t), \boldsymbol{r} \rangle - \frac{\alpha^2\Lambda}{2}\|\boldsymbol{r}\|^2. \tag{9}
$$

*Proof.*  First, we substitute $\boldsymbol{\delta} = -\alpha\boldsymbol{r}$. Then, according to the fundamental theorem of calculus we receive the equality:

$$
\mathcal{L}(\boldsymbol{\theta}_{t+1}) - \mathcal{L}(\boldsymbol{\theta}_t) = \mathcal{L}(\boldsymbol{\theta}_t + \boldsymbol{\delta}) - \mathcal{L}(\boldsymbol{\theta}_t) = \int_0^1 \langle \nabla\mathcal{L}(\boldsymbol{\theta}_t + s\boldsymbol{\delta}), \boldsymbol{\delta} \rangle \,\mathrm{d}s. \tag{10}
$$

Adding and substracting the value $\langle \nabla\mathcal{L}(\boldsymbol{\theta}_t), \delta \rangle = \int_0^1 \langle \nabla\mathcal{L}(\boldsymbol{\theta}_t), \delta \rangle \,\mathrm{d}s$ leads to:

$$
\mathcal{L}(\boldsymbol{\theta}_{t+1}) - \mathcal{L}(\boldsymbol{\theta}_t) = \langle \nabla\mathcal{L}(\boldsymbol{\theta}_t), \boldsymbol{\delta} \rangle + \int_0^1 \langle \nabla\mathcal{L}(\boldsymbol{\theta}_t + s\boldsymbol{\delta}) - \nabla\mathcal{L}(\boldsymbol{\theta}_t), \boldsymbol{\delta} \rangle \,\mathrm{d}s. \tag{11}
$$

Since the gradient satisfies the Lipschitz condition: $\|\nabla\mathcal{L}(\boldsymbol{\theta}_t + s\boldsymbol{\delta}) - \nabla\mathcal{L}(\boldsymbol{\theta}_t)\| \leq \Lambda\|\boldsymbol{\theta}_t + s\boldsymbol{\delta} - \boldsymbol{\theta}_t\|$ and due to inequality $\langle x, y \rangle \leq \|x\|\|y\|$, we can transform the integral, make the reverse substitution and get the final result:

$$
\mathcal{L}(\boldsymbol{\theta}_{t+1}) - \mathcal{L}(\boldsymbol{\theta}_t) \leq -\alpha\langle \nabla\mathcal{L}(\boldsymbol{\theta}_t), \boldsymbol{r} \rangle + \alpha^2\Lambda\|\boldsymbol{r}\|^2 \int_0^1 s \,\mathrm{d}s. \tag{12}
$$

$\square$

**Theorem 1.**  *Assume $\mathcal{L}^1(\boldsymbol{\theta}), \ldots, \mathcal{L}^T(\boldsymbol{\theta})$ are lower bounded continuously differentiable functions with Lipschitz continuous gradients with $\Lambda > 0$. Assume that the total loss function $\mathcal{L}(\boldsymbol{\theta})$ is a weighted sum of these functions with a pre-selected vector of preferences $\boldsymbol{w}$. Gradient descent with step size $\alpha \leq \frac{1}{\Lambda}$ and $\boldsymbol{\theta}$-aligned gradient converges if:*

    *1. $\boldsymbol{G}_{\boldsymbol{\theta}} = \boldsymbol{U}\boldsymbol{\Sigma}\boldsymbol{V}^\top$ (given by a SVD) and $\boldsymbol{\Sigma} = \operatorname{diag}\{\sigma_1, \ldots, \sigma_R\}$, where $\frac{\sigma_1}{\sigma_R} < C$.*

2. $\| \operatorname{proj}_{\mathcal{H}_V} \boldsymbol{w} \| \geq \varepsilon$, where $\mathcal{H}_V = \operatorname{span}(\boldsymbol{V})$.

*Proof.* Following the assumptions we find that the cumulative loss function is continuously differentiable and has a Lipschitz continuous gradient (denote this constant as $\Lambda$) and therefore it satisfies Lemma 1:

$$\mathcal{L}(\boldsymbol{\theta}_t) - \mathcal{L}(\boldsymbol{\theta}_{t+1}) \geq \alpha \langle \nabla_{\boldsymbol{\theta}} \mathcal{L}(\boldsymbol{\theta}_t), \nabla_{\boldsymbol{\theta}}^{\boldsymbol{\theta}a} \mathcal{L}(\boldsymbol{\theta}_t) \rangle - \frac{\alpha^2 \Lambda}{2} \left\| \nabla_{\boldsymbol{\theta}}^{\boldsymbol{\theta}a} \mathcal{L}(\boldsymbol{\theta}_t) \right\|^2. \tag{13}$$

By definition of $\boldsymbol{\theta}$-aligned gradient we get:

$$\langle \nabla_{\boldsymbol{\theta}} \mathcal{L}(\boldsymbol{\theta}_t), \nabla_{\boldsymbol{\theta}}^{\boldsymbol{\theta}a} \mathcal{L}(\boldsymbol{\theta}_t) \rangle = \sum_{r,r'=1}^{R} \sigma_r \sigma_R \boldsymbol{w}^\top \left( \boldsymbol{v}_r \boldsymbol{u}_r^\top \right) \left( \boldsymbol{u}_{r'} \boldsymbol{v}_{r'}^\top \right) \boldsymbol{w} = \sigma_R \sum_{r=1}^{R} \sigma_r (\boldsymbol{w}^\top \boldsymbol{v}_r)^2, \tag{14}$$

$$\langle \nabla_{\boldsymbol{\theta}}^{\boldsymbol{\theta}a} \mathcal{L}(\boldsymbol{\theta}_i), \nabla_{\boldsymbol{\theta}}^{\boldsymbol{\theta}a} \mathcal{L}(\boldsymbol{\theta}_i) \rangle = \sum_{r,r'=1}^{R} \sigma_R^2 \boldsymbol{w}^\top \left( \boldsymbol{v}_r \boldsymbol{u}_r^\top \right) \left( \boldsymbol{u}_{r'} \boldsymbol{v}_{r'}^\top \right) \boldsymbol{w} = \sigma_R^2 \sum_{r=1}^{R} (\boldsymbol{w}^\top \boldsymbol{v}_r)^2. \tag{15}$$

Since $\alpha \leq \frac{1}{\Lambda}$ and $\forall r : \frac{\sigma_r}{\sigma_R} > 1$, we can rewrite Eq. (13):

$$\mathcal{L}(\boldsymbol{\theta}_t) - \mathcal{L}(\boldsymbol{\theta}_{t+1}) \geq \sigma_R^2 \frac{\alpha}{2} \sum_{r=1}^{R} \underbrace{\left( 2\frac{\sigma_r}{\sigma_R} - 1 \right)}_{>1} \underbrace{\left( \boldsymbol{w}^\top \boldsymbol{v}_r \right)^2}_{=\|\boldsymbol{w}\|^2 \cos^2(\boldsymbol{w}, \boldsymbol{v}_r)} > \frac{\alpha \sigma_R^2}{2} \frac{\varepsilon^2}{\sigma_1^2} \sigma_1^2. \tag{16}$$

Following the assumption $\frac{\sigma_R}{\sigma_1} > C$. Moreover $\sigma_1 = \max_{\boldsymbol{x} \neq 0} \frac{\|\boldsymbol{G}_{\theta} \boldsymbol{x}\|}{\|\boldsymbol{x}\|}$, therefore $\sigma_1 \geq \frac{\|\nabla_{\boldsymbol{\theta}} \mathcal{L}(\boldsymbol{\theta}_t)\|}{\|\boldsymbol{w}\|}$. Then:

$$\mathcal{L}(\boldsymbol{\theta}_t) - \mathcal{L}(\boldsymbol{\theta}_{t+1}) > \frac{\alpha \varepsilon^2 C^2}{2\|\boldsymbol{w}\|^2} \|\nabla_{\boldsymbol{\theta}} \mathcal{L}(\boldsymbol{\theta}_t)\|^2. \tag{17}$$

The sequence of $\mathcal{L}(\boldsymbol{\theta}_t)$ is monotonically decreasing and bounded (under assumption) and therefore converges. Then $\mathcal{L}(\boldsymbol{\theta}_t) - \mathcal{L}(\boldsymbol{\theta}_{t+1}) \to 0$ if $t \to \infty$. Therefore, we have a local convergence of gradient descent:

$$\|\nabla_{\boldsymbol{\theta}} \mathcal{L}(\boldsymbol{\theta}_t)\|^2 < \frac{2\|\boldsymbol{w}\|^2}{\alpha C^2 \epsilon^2} \left( \mathcal{L}(\boldsymbol{\theta}_t) - \mathcal{L}(\boldsymbol{\theta}_{t+1}) \right) \to 0 \quad \text{as} \quad t \to \infty. \tag{18}$$

The same estimate appears in case of gradient descent. Therefore, the convergence of the proposed method is similar to that of gradient descent. $\square$

**Theorem 2.** *Assume $\mathcal{L}^1(\boldsymbol{\theta}), \ldots, \mathcal{L}^T(\boldsymbol{\theta})$ are lower bounded continuously differentiable functions with Lipschitz continuous gradients with $\Lambda > 0$. Assume that the total loss function $\mathcal{L}(\boldsymbol{\theta})$ is a weighted sum of these functions with a pre-selected vector of preferences $\boldsymbol{w}$. The gradient descent with step size $\alpha \leq \frac{1}{\Lambda}$ and a $\boldsymbol{Z}$-aligned gradient converges if:*

1. *$\boldsymbol{G}_{\boldsymbol{Z}} = \boldsymbol{U} \boldsymbol{\Sigma} \boldsymbol{V}^\top$ (given by a SVD) and $\boldsymbol{\Sigma} = \operatorname{diag}\{\sigma_1, \ldots, \sigma_R\}$, where $\frac{\sigma_1}{\sigma_R} < C_\sigma$.*

2. *$\| \operatorname{proj}_{\mathcal{H}_V} \boldsymbol{w} \| \geq \varepsilon$, where $\mathcal{H}_V = \operatorname{span}(\boldsymbol{V})$.*

3. *$\boldsymbol{J} = \frac{\partial \boldsymbol{Z}}{\partial \theta}$ is full rank.*

4. *$\boldsymbol{J} = \boldsymbol{P} \boldsymbol{S} \boldsymbol{Q}^\top$ (given by a SVD) and $\boldsymbol{S} = \operatorname{diag}\{\lambda_1, \ldots, \lambda_K\}$, where $\frac{\lambda_1}{\lambda_K} < C_\lambda$.*

*Proof.* Following the assumptions we find that the cumulative loss function is continuously differentiable and has a Lipschitz continuous gradient (with constant $\Lambda > 0$) and therefore it satisfies Lemma 1:

$$\mathcal{L}(\boldsymbol{\theta}_t) - \mathcal{L}(\boldsymbol{\theta}_{t+1}) \geq \alpha \langle \nabla_{\boldsymbol{\theta}} \mathcal{L}(\boldsymbol{\theta}_t), \nabla_{\boldsymbol{\theta}}^{\boldsymbol{Z}a} \mathcal{L}(\boldsymbol{\theta}) \rangle - \frac{\alpha^2 \Lambda}{2} \|\nabla_{\boldsymbol{\theta}}^{\boldsymbol{Z}a} \mathcal{L}(\boldsymbol{\theta})\|^2. \tag{19}$$

Since $\alpha \leq \frac{1}{\Lambda}$, we can rewrite this inequality:

$$\mathcal{L}(\boldsymbol{\theta}_t) - \mathcal{L}(\boldsymbol{\theta}_{t+1}) \geq \frac{\alpha}{2} \langle 2\nabla_{\boldsymbol{\theta}} \mathcal{L}(\boldsymbol{\theta}_t) - \nabla_{\boldsymbol{\theta}}^{\boldsymbol{Z}a} \mathcal{L}(\boldsymbol{\theta}_t), \nabla_{\boldsymbol{\theta}}^{\boldsymbol{Z}a} \mathcal{L}(\boldsymbol{\theta}_t) \rangle. \tag{20}$$

Consider the left vector of a dot product in the right part of Eq. (20). By definition of $\boldsymbol{Z}$-aligned gradient we get:

$$\nabla_{\boldsymbol{\theta}}^{\boldsymbol{Z}a} \mathcal{L}(\boldsymbol{\theta}_t) = \sum_{r=1}^{R} \sigma_R (\boldsymbol{w}^\top \boldsymbol{v}_r) \boldsymbol{J} \boldsymbol{u}_r, \tag{21}$$

$$2\nabla_{\boldsymbol{\theta}} \mathcal{L}(\boldsymbol{\theta}_t) - \nabla_{\boldsymbol{\theta}}^{\boldsymbol{Z}a} \mathcal{L}(\boldsymbol{\theta}_t) = \sum_{r=1}^{R} (2\sigma_r - \sigma_R)(\boldsymbol{w}^\top \boldsymbol{v}_r) \boldsymbol{J} \boldsymbol{u}_r. \tag{22}$$

Following the assumption $\boldsymbol{J}^\top \boldsymbol{J}$ is positive definite. Any positive definite matrix is congruent to a diagonal with positive and ordered eigenvalues on the main diagonal. Thus replacing all eigenvalues $\lambda_i^2$ with the smallest one $\lambda_K^2$ does not increase the right side of Eq. (19) resulting in the inequality:

$$\langle 2\nabla_{\boldsymbol{\theta}} \mathcal{L}(\boldsymbol{\theta}_t) - \nabla_{\boldsymbol{\theta}}^{\boldsymbol{Z}a} \mathcal{L}(\boldsymbol{\theta}_t), \nabla_{\boldsymbol{\theta}}^{\boldsymbol{Z}a} \mathcal{L}(\boldsymbol{\theta}_t) \rangle \geq \sigma_R^2 \lambda_K^2 \sum_{r=1}^{R} \underbrace{\left( 2\frac{\sigma_r}{\sigma_R} - 1 \right)}_{>1} \underbrace{\left( \boldsymbol{w}^\top \boldsymbol{v}_r \right)^2}_{=\|\boldsymbol{w}\|^2 \cos^2(\boldsymbol{w}, \boldsymbol{v}_r)}. \tag{23}$$

Therefore:

$$\mathcal{L}(\boldsymbol{\theta}_t) - \mathcal{L}(\boldsymbol{\theta}_{t+1}) \geq \frac{\alpha \varepsilon^2 \sigma_R^2 \lambda_K^2}{2\sigma_1^2 \lambda_1^2} \sigma_1^2 \lambda_1^2. \tag{24}$$

Following the assumption $\frac{\sigma_R}{\sigma_1} > C_\sigma$ and $\frac{\lambda_K}{\lambda_1} > C_\lambda$. Moreover $\sigma_1 = \max_{\boldsymbol{x} \neq 0} \frac{\|\boldsymbol{G}_{\boldsymbol{Z}} \boldsymbol{x}\|}{\|\boldsymbol{x}\|} \geq \frac{\|\nabla_{\boldsymbol{Z}} \mathcal{L}(\boldsymbol{\theta})\|}{\|\boldsymbol{w}\|}$ and $\lambda_1 = \|\boldsymbol{J}\|$. Therefore, we get the final bound:

$$\mathcal{L}(\boldsymbol{\theta}_t) - \mathcal{L}(\boldsymbol{\theta}_{t+1}) \geq \frac{\alpha \varepsilon^2 C_\sigma^2 C_\lambda^2}{2\|\boldsymbol{w}\|^2} \|\boldsymbol{G}_{\boldsymbol{Z}} \boldsymbol{w}\|^2 \|\boldsymbol{J}\|^2 \geq \frac{\alpha \varepsilon^2 C_\sigma^2 C_\lambda^2}{2\|\boldsymbol{w}\|^2} \|\nabla_{\boldsymbol{\theta}} \mathcal{L}(\boldsymbol{\theta}_t)\|^2. \tag{25}$$

The sequence of $\mathcal{L}(\boldsymbol{\theta}_t)$ is monotonically decreasing and bounded (under assumption) and therefore converges. Then $\mathcal{L}(\boldsymbol{\theta}_t) - \mathcal{L}(\boldsymbol{\theta}_{t+1}) \to 0$ if $t \to \infty$. Therefore, we have local convergence of gradient descent:

$$\|\nabla_{\boldsymbol{\theta}} \mathcal{L}(\boldsymbol{\theta}_t)\|^2 < \frac{2\|\boldsymbol{w}\|^2}{\alpha C_\sigma^2 C_\lambda^2 \varepsilon^2} \left( \mathcal{L}(\boldsymbol{\theta}_t) - \mathcal{L}(\boldsymbol{\theta}_{t+1}) \right) \to 0 \quad \text{as} \quad t \to \infty. \tag{26}$$

$\square$

## C   METHODS WITH MOMENTUM

**Definition 4.** *A function $\mathcal{L}(x) : \mathcal{X} \to \mathbb{R}$ is $\mu$-strongly convex if for any $x, y \in \mathcal{X}$*

$$\mathcal{L}(y) \geq \mathcal{L}(x) + \langle \nabla \mathcal{L}(x), x - y \rangle + \frac{\mu}{2} \|x - y\|^2. \tag{27}$$

**Definition 5.** *A continuously differentiable function $\mathcal{L}(x) : \mathcal{X} \to \mathbb{R}$ is $\Lambda$-smooth if for any $x, y \in \mathcal{X}$:*

$$\|\nabla \mathcal{L}(x) - \nabla \mathcal{L}(y)\| \leq \Lambda \|x - y\|. \tag{28}$$

**Lemma 2.** *A $\boldsymbol{\theta}$-aligned and $\boldsymbol{Z}$-aligned gradient can be expressed as:*

$$\nabla_{\boldsymbol{\theta}}^{*a} \mathcal{L}(\boldsymbol{\theta}_t) = \sum_{i=1}^{T} \left( \sum_{k=1}^{T} w_k \left[ \sum_{r=1}^{R} \frac{\sigma_R}{\sigma_r} v_r^i v_r^k \right] \right) \nabla_{\boldsymbol{\theta}} \mathcal{L}^i(\boldsymbol{\theta}_t). \tag{29}$$

*Proof.* We consider only the $\boldsymbol{Z}$-aligned case. The $\boldsymbol{\theta}$-aligned case may be proved using analogous logic. According to SVD theorem, $\boldsymbol{U} = \boldsymbol{G}_{\boldsymbol{Z}} \boldsymbol{V} \Sigma^{-1}$ in matrix form that in vector form turns into $\boldsymbol{u}_r = \frac{1}{\sigma_r} \boldsymbol{G}_{\boldsymbol{Z}} \boldsymbol{v}_r$. Then, by definition of a $\boldsymbol{Z}$-aligned gradient:

$$\sigma_R \hat{\boldsymbol{G}}_{\boldsymbol{Z}_t} = \sigma_R \sum_{r=1}^{R} \boldsymbol{u}_r \boldsymbol{v}_r^\top = \boldsymbol{G}_{\boldsymbol{Z}_t} \sum_{r=1}^{R} \frac{\sigma_R}{\sigma_r} \boldsymbol{v}_r \boldsymbol{v}_r^\top. \tag{30}$$

If we rewrite this dot product, then we get:

$$\sigma_R \hat{\boldsymbol{G}}_{\boldsymbol{Z}_t} = \left( \sum_{i=1}^{T} \underbrace{\left[ \sum_{r=1}^{R} \frac{\sigma_R}{\sigma_r} v_r^i v_r^1 \right]}_{a_{i1}} \nabla_{\boldsymbol{Z}} \mathcal{L}^i(\boldsymbol{\theta}_t), \ldots, \sum_{i=1}^{T} \underbrace{\left[ \sum_{r=1}^{R} \frac{\sigma_R}{\sigma_r} v_r^i v_r^T \right]}_{a_{iT}} \nabla_{\boldsymbol{Z}} \mathcal{L}^i(\boldsymbol{\theta}_t) \right). \qquad (31)$$

Since $\sigma_R \boldsymbol{J} \hat{\boldsymbol{G}}_{\boldsymbol{Z}_t} = \left( \ldots, \sum a_{ik} \boldsymbol{J} \nabla_{\boldsymbol{Z}} \mathcal{L}^i(\boldsymbol{\theta}_t), \ldots \right) = \left( \ldots, \sum a_{ik} \nabla_{\boldsymbol{\theta}} \mathcal{L}^i(\boldsymbol{\theta}_t), \ldots \right)$, then the final statement follows from the definition of the $\boldsymbol{Z}$-aligned gradient:

$$\nabla_{\boldsymbol{\theta}}^{\boldsymbol{Z}a} \mathcal{L}(\boldsymbol{\theta}_t) = \sigma_R \boldsymbol{J} \hat{\boldsymbol{G}}_{\boldsymbol{Z}_t} \boldsymbol{w} = \sum_{k=1}^{T} \sum_{i=1}^{T} w_k a_{ik} \nabla_{\boldsymbol{\theta}} \mathcal{L}^i(\boldsymbol{\theta}_t) = \sum_{i=1}^{T} \left( \sum_{k=1}^{T} w_k a_{ik} \right) \nabla_{\boldsymbol{\theta}} \mathcal{L}^i(\boldsymbol{\theta}_t). \qquad (32)$$

Similarly, this equation can be derived for $\boldsymbol{\theta}$-aligned case:

$$\nabla_{\boldsymbol{\theta}}^{\boldsymbol{\theta}a} \mathcal{L}(\boldsymbol{\theta}_t) = \sigma_R \hat{\boldsymbol{G}}_{\boldsymbol{\theta}_t} \boldsymbol{w} = \sum_{k=1}^{T} \sum_{i=1}^{T} w_k \hat{a}_{ik} \nabla_{\boldsymbol{\theta}} \mathcal{L}^i(\boldsymbol{\theta}_t) = \sum_{i=1}^{T} \left( \sum_{k=1}^{T} w_k \hat{a}_{ik} \right) \nabla_{\boldsymbol{\theta}} \mathcal{L}^i(\boldsymbol{\theta}_t). \qquad (33)$$

$\square$

**Theorem 3.** *Assume $\mathcal{L}^1(\boldsymbol{\theta}), \ldots, \mathcal{L}^T(\boldsymbol{\theta})$ are strongly convex with constant $\mu_i > 0$ and smooth with constant $\Lambda_i > 0$. Assume $\hat{\mu}_t = \sum_{i=1}^{T} \left( \sum_{k=1}^{T} w_k a_{ik} \right) \mu_i$ and $\hat{\Lambda}_t = \sum_{i=1}^{T} \left( \sum_{k=1}^{T} w_k a_{ik} \right) \Lambda_i$. The heavy ball method with $\alpha_t = \frac{4}{(\sqrt{\hat{\Lambda}_t} + \sqrt{\hat{\mu}_t})^2}$ and $\beta_t = \max\{|1 - \sqrt{\alpha_t \hat{\mu}_t}|, |1 - \sqrt{\alpha_t \hat{\Lambda}_t}|\}$ and the $\boldsymbol{\theta}$-aligned or $\boldsymbol{Z}$-aligned gradient will converge linearly to an optimal value $\mathcal{L}(\boldsymbol{\theta}^*)$ if:*

1. *($\boldsymbol{\theta}$-aligned) $\boldsymbol{G}_{\boldsymbol{\theta}} = \boldsymbol{U} \boldsymbol{\Sigma} \boldsymbol{V}^\top$ (given by a SVD) and $\boldsymbol{\Sigma} = diag\{\sigma_1, \ldots, \sigma_R\}$, where $\frac{\sigma_R}{\sigma_1} > C_\sigma$.*

2. *($\boldsymbol{Z}$-aligned) $\boldsymbol{G}_{\boldsymbol{Z}} = \boldsymbol{U} \boldsymbol{\Sigma} \boldsymbol{V}^\top$ (given by a SVD) and $\boldsymbol{\Sigma} = diag\{\sigma_1, \ldots, \sigma_R\}$, where $\frac{\sigma_R}{\sigma_1} > C_\sigma$.*

3. *($\boldsymbol{Z}$-aligned) $\boldsymbol{J}$ is full rank.*

4. *(both) $\|\operatorname{proj}_{\mathcal{H}_{\boldsymbol{V}}} \boldsymbol{w}\| \geq \varepsilon$, where $\mathcal{H}_{\boldsymbol{V}} = \operatorname{span}(\boldsymbol{V})$.*

*Proof.* The conditions of the theorem ensure the existence of finite and non-zero $\boldsymbol{\theta}$ and $\boldsymbol{Z}$-aligned gradients. Due to existence Lemma 2 is valid. We will not separate $\boldsymbol{\theta}$-alignment and $\boldsymbol{Z}$-alignment in this theorem due to similarity of the proving procedure. Instead we denote $\nabla_{\boldsymbol{\theta}}^{*a} \mathcal{L}(\boldsymbol{\theta}_t)$ as a general aligned gradient.

Consider the convergence of a heavy ball method using matrix notation following Sun (2015). The next point is computed by definition, *i.e.* $\boldsymbol{\theta}_{t+1} = \boldsymbol{\theta}_t - \alpha_t \nabla_{\boldsymbol{\theta}}^{*a} \mathcal{L}(\boldsymbol{\theta}_t) + \beta_t (\boldsymbol{\theta}_t - \boldsymbol{\theta}_{t-1})$. Then we can estimate the squared discrepancy between the current solution and the optimal point using the explicit expression for aligned gradient from Lemma 2:

$$\left\| \begin{bmatrix} \boldsymbol{\theta}_{t+1} - \boldsymbol{\theta}^* \\ \boldsymbol{\theta}_t - \boldsymbol{\theta}^* \end{bmatrix} \right\|_2 = \left\| \begin{bmatrix} \boldsymbol{\theta}_t + \beta_t(\boldsymbol{\theta}_t - \boldsymbol{\theta}_{t-1}) - \boldsymbol{\theta}^* \\ \boldsymbol{\theta}_t - \boldsymbol{\theta}^* \end{bmatrix} - \alpha_t \begin{bmatrix} \nabla_{\boldsymbol{\theta}}^{*a} \mathcal{L}(\boldsymbol{\theta}_t) \\ 0 \end{bmatrix} \right\|_2$$

$$= \left\| \begin{bmatrix} \boldsymbol{\theta}_t + \beta_t(\boldsymbol{\theta}_t - \boldsymbol{\theta}_{t-1}) - \boldsymbol{\theta}^* \\ \boldsymbol{\theta}_t - \boldsymbol{\theta}^* \end{bmatrix} - \alpha_t \begin{bmatrix} \sum_{k=1}^{T} w_k' \nabla_{\boldsymbol{\theta}} \mathcal{L}^k(\boldsymbol{\theta}_t) \\ 0 \end{bmatrix} \right\|_2, \qquad (34)$$

Note that for all gradients for some point $\boldsymbol{\Theta}_t^k$ on the line segment between $\boldsymbol{\theta}_t$ and $\boldsymbol{\theta}^*$ the following expression is true: $\nabla_{\boldsymbol{\theta}} \mathcal{L}^k(\boldsymbol{\theta}_t) = \nabla_{\boldsymbol{\theta}}^2 \mathcal{L}^k(\boldsymbol{\Theta}_t^k)(\boldsymbol{\theta}_t - \boldsymbol{\theta}^*)$, where $\nabla_{\boldsymbol{\theta}}^2 \mathcal{L}^k(\boldsymbol{\Theta}_t^k)$ is a Hessian of a function $\mathcal{L}^k$. Denote $\boldsymbol{H}_t = \sum_{k=1}^{T} w_k' \nabla_{\boldsymbol{\theta}}^2 \mathcal{L}^k(\boldsymbol{\Theta}_t^k)$. Hence, $\boldsymbol{H}_t$ is a Hessian of a total loss function. Then we

get:

$$\left\|\begin{bmatrix}\boldsymbol{\theta}_{t+1}-\boldsymbol{\theta}^*\\\boldsymbol{\theta}_t-\boldsymbol{\theta}^*\end{bmatrix}\right\|_2 = \left\|\begin{bmatrix}\boldsymbol{\theta}_t+\beta_t(\boldsymbol{\theta}_t-\boldsymbol{\theta}_{t-1})-\boldsymbol{\theta}^*\\\boldsymbol{\theta}_t-\boldsymbol{\theta}^*\end{bmatrix}-\alpha_t\begin{bmatrix}\boldsymbol{H}_t(\boldsymbol{\theta}_t-\boldsymbol{\theta}^*)\\0\end{bmatrix}\right\|_2$$

$$= \left\|\begin{bmatrix}(1+\beta_t)I-\alpha_t\boldsymbol{H}_t & -\beta_t I\\I & 0\end{bmatrix}\begin{bmatrix}\boldsymbol{\theta}_t-\boldsymbol{\theta}^*\\\boldsymbol{\theta}_{t-1}-\boldsymbol{\theta}^*\end{bmatrix}\right\|_2$$

$$\le \left\|\begin{bmatrix}(1+\beta_t)I-\alpha_t\boldsymbol{H}_t & -\beta_t I\\I & 0\end{bmatrix}\right\|_2\left\|\begin{bmatrix}\boldsymbol{\theta}_t-\boldsymbol{\theta}^*\\\boldsymbol{\theta}_{t-1}-\boldsymbol{\theta}^*\end{bmatrix}\right\|_2. \tag{35}$$

By strong convexity and smoothness of $\mathcal{L}^k(\boldsymbol{\theta})$ the eigenvalues of $\nabla_{\boldsymbol{\theta}}^2\mathcal{L}^k(\Theta_t^k)$ are in the interval $[\mu_k,\Lambda_k]$. Hence eigenvalues of $\boldsymbol{H}_t$ are between $\hat{\mu}_t$ and $\hat{\Lambda}_t$. Therefore, following Lemma 3.1 introduced in Sun (2015), the $L_2$ norm of a matrix in the right side of Eq. (35) is bounded:

$$\left\|\begin{bmatrix}(1+\beta_t)I-\alpha_t\boldsymbol{H}_t & -\beta_t I\\I & 0\end{bmatrix}\right\|_2 \le \max\left\{|1-\sqrt{\alpha_t\hat{\mu}_t}|,|1-\sqrt{\alpha_t\hat{\Lambda}_t}|\right\}. \tag{36}$$

Now, we can plug in step size $\alpha_t=\frac{4}{(\sqrt{\hat{\Lambda}_t}+\sqrt{\hat{\mu}_t})^2}$ and Eq. (36) into Eq. (35):

$$\left\|\begin{bmatrix}\boldsymbol{\theta}_{t+1}-\boldsymbol{\theta}^*\\\boldsymbol{\theta}_t-\boldsymbol{\theta}^*\end{bmatrix}\right\|_2 \le \frac{\sqrt{\gamma_t}-1}{\sqrt{\gamma_t}+1}\left\|\begin{bmatrix}\boldsymbol{\theta}_t-\boldsymbol{\theta}^*\\\boldsymbol{\theta}_{t-1}-\boldsymbol{\theta}^*\end{bmatrix}\right\|_2 \le \left\|\begin{bmatrix}\boldsymbol{\theta}_t-\boldsymbol{\theta}^*\\\boldsymbol{\theta}_{t-1}-\boldsymbol{\theta}^*\end{bmatrix}\right\|_2, \tag{37}$$

where $\gamma_t=\frac{\hat{\Lambda}_t}{\hat{\mu}_t}$. Hence the heavy ball method with this gradient will converge linearly to an optimal value $\mathcal{L}(\boldsymbol{\theta}^*)$.

$\square$

# D   IMPLEMENTATION DETAILS

In this section, we provide additional experimental results and more details on the training pipelines. The proposed approach and all baseline methods are implemented in PyTorch (Paszke et al., 2019).

## D.1   MULTI-LABEL CLASSIFICATION

In addition to the radar chart presented in the main text, we present more detailed results on the CELEBA data set here. Specifically, we report the binary classification accuracy of each attribute for the proposed methods and the baselines on the validation split. The results are summarized in Table 5. Following Sener & Koltun (2018) we use ResNet-18 with 40 separate fully-connected (FC) layers as task-specific functions. Table 7 shows the architecture used in the experiments in more detail. The images of the data set are resized to $64\times64\times3$ as proposed in Sener & Koltun (2018). We empirically found that learning rate $10^{-3}$ gives better results. The learning rate is gradually decreased during training. All models are optimized using Adam solver and trained for 20 epochs with batch size 256.

## D.2   CAMERA RELOCALIZATION

For image-based localization, we follow Kendall et al. (2015) and adapt ResNet-34 architecture by removing the last FC layer but keep the convolutional and pooling layers intact. As a decoder, we use two separate FC layers with 3 and 4 outputs to regress camera orientation and translation, respectively. The network architecture is presented in Table 7. The 7SCENES (Shotton et al., 2013) data set is used for training and evaluation. The data set consists of $640\times480$ RGB-D images captured by a Kinect device and cover 7 different indoor scenes. As a preprocessing step, the images are re-scaled in such a way that the smaller image side is 256 pixels. All models were trained on random crops of $224\times224$ pixels and evaluated on central crops. We use batch size 128 and train for 120 epochs. The learning rate is initialized to $10^{-3}$ and decreased by 10 every 40 epochs. The Adam solver is used for optimization. Each method is trained 10 times with different random seeds for every scene and average performance is provided in Table 6.

Table 5: CELEBA performance. Following Sener & Koltun (2018), we report accuracy on the validation split. The best score is in **bold** and the second best score is underlined.

| | MGDA | MGDA-UB | GradNorm | PCGrad | Uncertainty | $\theta$-aligned (ours) | $Z$-aligned (ours) |
|---|---|---|---|---|---|---|---|
| | | | | MTL methods | | | |
| **Easy attributes** | | | | | | | |
| Attr. 0 | 92.39 | 93.05 | 93.09 | **94.02** | 93.96 | 93.51 | 93.69 |
| Attr. 4 | 98.79 | 98.74 | 98.69 | 98.80 | 98.77 | 98.77 | **98.84** |
| Attr. 5 | 94.51 | 95.47 | 95.30 | **95.72** | 95.42 | 95.52 | 95.36 |
| Attr. 9 | 93.77 | 94.93 | 95.22 | **95.45** | 95.16 | 95.39 | 95.38 |
| Attr. 10 | 96.12 | 96.12 | 96.20 | **96.40** | 96.07 | 95.77 | 96.04 |
| Attr. 13 | 95.14 | 95.05 | 95.33 | **95.36** | 95.17 | **95.36** | 95.28 |
| Attr. 14 | 96.03 | 96.22 | 96.09 | **96.43** | 96.24 | 96.42 | 96.23 |
| Attr. 16 | 95.88 | 96.18 | 96.18 | 96.40 | 96.39 | 96.13 | **96.41** |
| Attr. 17 | 97.27 | 97.67 | 97.19 | **97.91** | 97.76 | 97.70 | 97.78 |
| Attr. 20 | 97.25 | 98.01 | 98.13 | **98.53** | 98.52 | 98.02 | 98.41 |
| Attr. 21 | 89.83 | 92.60 | 93.01 | **93.72** | 93.57 | 93.54 | 93.32 |
| Attr. 22 | 96.13 | 95.91 | 95.60 | **96.20** | 95.99 | 96.11 | 96.05 |
| Attr. 24 | 94.62 | 95.00 | 95.15 | **95.75** | 95.64 | 95.65 | 95.46 |
| Attr. 26 | 96.54 | 96.67 | 96.44 | **96.79** | 96.71 | 96.74 | 96.71 |
| Attr. 28 | 94.18 | 94.42 | 94.40 | **94.60** | 94.47 | 94.50 | 94.43 |
| Attr. 29 | 94.60 | 94.73 | 94.63 | 95.14 | 95.11 | **95.17** | 95.02 |
| Attr. 30 | 96.00 | 96.56 | 95.91 | 96.79 | 96.86 | **97.02** | 96.94 |
| Attr. 35 | 98.70 | **98.92** | 98.69 | 98.90 | 98.73 | 98.84 | 98.87 |
| Attr. 38 | 96.19 | 96.13 | 95.71 | **96.51** | 96.33 | 96.00 | 96.34 |
| **Hard attributes** | | | | | | | |
| Attr. 1 | 82.09 | 82.67 | 82.08 | **84.91** | 84.27 | **84.91** | 84.79 |
| Attr. 2 | 79.94 | 80.14 | 81.26 | **81.36** | 80.50 | 80.61 | 81.30 |
| Attr. 3 | 82.89 | 82.86 | 83.87 | **84.06** | 83.45 | 83.74 | 83.75 |
| Attr. 6 | 85.73 | 85.66 | **86.09** | 83.43 | 85.92 | 82.22 | 84.06 |
| Attr. 7 | 81.76 | 81.84 | 82.68 | 82.70 | 82.48 | **82.84** | 82.56 |
| Attr. 8 | 89.48 | 90.67 | 91.35 | **91.42** | 90.45 | 89.45 | 91.28 |
| Attr. 11 | 81.95 | 84.32 | 84.89 | **85.58** | 85.11 | 83.73 | 85.31 |
| Attr. 12 | 91.83 | 92.19 | 92.06 | **92.54** | 92.32 | 92.19 | 92.46 |
| Attr. 18 | 90.64 | 91.02 | 91.77 | **92.14** | 90.82 | 91.21 | 91.67 |
| Attr. 19 | 86.60 | 87.48 | 87.94 | **88.58** | 87.65 | 88.28 | 87.96 |
| Attr. 23 | 93.39 | 92.87 | 92.29 | 92.75 | 92.89 | **93.41** | 92.36 |
| Attr. 25 | 75.60 | 75.14 | 75.57 | **76.14** | 75.87 | 76.04 | 75.84 |
| Attr. 27 | 75.82 | 75.72 | 77.26 | **77.62** | 76.43 | 77.25 | 77.24 |
| Attr. 31 | 91.17 | 92.15 | 92.65 | **93.04** | 92.99 | 92.85 | 92.68 |
| Attr. 32 | 80.66 | 81.54 | 82.99 | **83.44** | 83.11 | 82.97 | 83.34 |
| Attr. 33 | 81.42 | 83.27 | **85.95** | 85.53 | 85.66 | 84.56 | 85.63 |
| Attr. 34 | 86.37 | 87.29 | 89.55 | **90.48** | 89.17 | 90.31 | 90.32 |
| Attr. 36 | 91.30 | 91.56 | 92.71 | 92.61 | 92.69 | 92.14 | **92.86** |
| Attr. 37 | 88.35 | 88.15 | 88.25 | **88.90** | 87.68 | 88.82 | 88.38 |
| Attr. 39 | 85.67 | 86.03 | 86.93 | **88.13** | 86.94 | 87.07 | 87.58 |

Table 6: Camera relocalization performance of the proposed method and existing MTL approaches for the 7SCENES data set. We report translation and orientation accuracy in terms of meters and degrees, respectively. The reported results are mean±std over 10 random seeds.

(a) Translation performance (meters).

| | CHESS | FIRE | HEADS | Scenes OFFICE | PUMPKIN | KITCHEN | STAIRS | Mean |
|---|---|---|---|---|---|---|---|---|
| uniform | 0.49±0.01 | 0.91±0.21 | 1.03±0.44 | 0.68±0.02 | 0.70±0.01 | 0.90±0.02 | 0.55±0.06 | 0.75±0.07 |
| MGDA | 0.32±0.01 | 0.65±0.07 | 0.75±0.19 | 0.31±0.01 | 0.44±0.02 | 0.34±0.01 | 0.50±0.02 | 0.47±0.03 |
| MGDA-UB | 0.33±0.01 | 0.73±0.13 | 0.72±0.16 | 0.58±0.03 | 0.48±0.03 | 0.64±0.03 | 0.52±0.02 | 0.57±0.03 |
| GradNorm | 0.18±0.01 | 0.33±0.01 | 0.25±0.01 | 0.25±0.00 | 0.25±0.00 | 0.27±0.01 | 0.43±0.02 | 0.28±0.00 |
| PCGrad | 0.17±0.00 | 0.33±0.01 | 0.24±0.01 | 0.24±0.01 | 0.25±0.01 | 0.26±0.01 | 0.40±0.03 | 0.27±0.01 |
| Uncertainty | 0.19±0.01 | 0.37±0.02 | 0.29±0.02 | 0.26±0.00 | 0.26±0.01 | 0.29±0.01 | 0.46±0.02 | 0.30±0.00 |
| $\theta$-aligned | 0.26±0.02 | 0.53±0.03 | 0.63±0.03 | 0.26±0.00 | 0.35±0.02 | 0.29±0.00 | 0.59±0.03 | 0.42±0.01 |
| $Z$-aligned | 0.20±0.03 | 0.48±0.04 | 0.50±0.06 | 0.29±0.01 | 0.31±0.02 | 0.32±0.00 | 0.58±0.07 | 0.38±0.02 |

(b) Orientation performance (degrees).

| | CHESS | FIRE | HEADS | Scenes OFFICE | PUMPKIN | KITCHEN | STAIRS | Mean |
|---|---|---|---|---|---|---|---|---|
| uniform | 6.26±0.20 | 11.75±0.24 | 13.08±0.13 | 8.26±0.16 | 6.68±0.15 | 8.72±0.21 | 11.81±0.24 | 9.51±0.07 |
| MGDA | 6.02±0.10 | 11.59±0.29 | 12.99±0.17 | 8.55±0.15 | 6.59±0.09 | 8.84±0.11 | 11.61±0.18 | 9.46±0.06 |
| MGDA-UB | 5.95±0.09 | 11.76±0.29 | 12.96±0.16 | 8.44±0.2 | 6.42±0.10 | 8.73±0.18 | 11.58±0.31 | 9.40±0.07 |
| GradNorm | 7.41±0.14 | 12.77±0.28 | 14.35±0.48 | 9.76±0.2 | 8.75±0.34 | 9.74±0.24 | 13.94±0.30 | 10.96±0.10 |
| PCGrad | 8.29±0.21 | 13.56±0.34 | 14.96±0.59 | 10.81±0.18 | 9.86±0.36 | 10.43±0.19 | 14.42±0.34 | 11.76±0.15 |
| Uncertainty | 9.04±0.35 | 15.44±0.77 | 18.69±1.08 | 10.60±0.32 | 10.19±0.48 | 10.35±0.13 | 15.69±0.97 | 12.86±0.29 |
| $\theta$-aligned | 7.47±0.49 | 14.40±1.36 | 18.95±1.30 | 9.14±0.25 | 7.72±0.67 | 9.58±0.21 | 14.08±0.57 | 11.62±0.31 |
| $Z$-aligned | 5.81±0.19 | 12.18±0.68 | 13.58±0.68 | 9.03±0.17 | 6.31±0.16 | 9.38±0.15 | 12.26±0.32 | 9.79±0.17 |

Table 7: The network architectures used in our experiments for (1) Multi-label classification (CELEBA), (2) camera relocalization (7SCENES), (3) Scene understanding (CITYSCAPES), and (4) Mulati-task classification (MULTIMNIST). Notation: PPM stands for Pyramid Pooling Module (Zhao et al., 2017); DE is depth estimation; IS and SS are instance and semantic segmentation respectively.

| | CELEBA ResNet-18 | 7SCENES ResNet-34 | CITYSCAPES ResNet-50 | MULTIMNIST LeNet |
|---|---|---|---|---|
| | Conv, 3×3, 64, s=1 | Conv, 7×7, 64, s=2  MaxPool, 3×3, s=2 | | |
| encoder | $\begin{bmatrix} 3\times3, 64 \\ 3\times3, 64 \end{bmatrix} \times 2$ | $\begin{bmatrix} 3\times3, 64 \\ 3\times3, 64 \end{bmatrix} \times 3$ | $\begin{bmatrix} 1\times1, 64 \\ 3\times3, 64 \\ 1\times1, 256 \end{bmatrix} \times 3$ | Conv, 5×5, 10  MaxPool, 2×2  Conv, 5×5, 20  MaxPool, 2×2  FC, 50 |
| | $\begin{bmatrix} 3\times3, 128 \\ 3\times3, 128 \end{bmatrix} \times 2$ | $\begin{bmatrix} 3\times3, 128 \\ 3\times3, 128 \end{bmatrix} \times 4$ | $\begin{bmatrix} 1\times1, 128 \\ 3\times3, 128 \\ 1\times1, 512 \end{bmatrix} \times 4$ | |
| | $\begin{bmatrix} 3\times3, 256 \\ 3\times3, 256 \end{bmatrix} \times 2$ | $\begin{bmatrix} 3\times3, 256 \\ 3\times3, 256 \end{bmatrix} \times 6$ | $\begin{bmatrix} 1\times1, 256 \\ 3\times3, 256 \\ 1\times1, 1024 \end{bmatrix} \times 6$ | |
| | $\begin{bmatrix} 3\times3, 512 \\ 3\times3, 512 \end{bmatrix} \times 2$ | $\begin{bmatrix} 3\times3, 512 \\ 3\times3, 512 \end{bmatrix} \times 3$ | $\begin{bmatrix} 1\times1, 512 \\ 3\times3, 512 \\ 1\times1, 2048 \end{bmatrix} \times 3$ | |
| decoder | (FC, 2)×40 | FC, 3 (Translation)  FC, 4 (Orientation) | PPM(1) (DE)  PPM(2) (IS)  PPM(19) (SS) | FC, 10 (Left)  FC, 10 (Right) |

