# OpenReview forum: "Independent Component Alignment for Multi-task Learning"
_ICLR.cc/2022/Conference — ICLR 2022 Submitted_

### Official Review · Reviewer_RbSw · 2021-11-01

**Correctness:** 4
**Technical Novelty And Significance:** 3
**Empirical Novelty And Significance:** 3
**Recommendation:** 6
**Confidence:** 3

**Main Review:**

Strengths:
-	The idea of balancing multi-task training by aligning the independent components of the gradient of the training objective makes sense to me.
-	The proposed method achieves favorable performance on multiple multi-task learning benchmarks.
-	The paper is well written and easy to follow.

Weaknesses:
- The performance on some benchmarks are not that good, such as the results on CELEBA,



**Summary Of The Paper:**

This paper proposes a gradient-based multi-task learning method to balance multi-task training by aligning the independent components of the training objective. Experiments conducted on several MTL problems are given. The method is claimed to be scalable, robust to overfitting, and able to seamlessly handle multi-task objectives with a large difference in gradient magnitudes.

**Summary Of The Review:**

The idea of this paper is interesting and makes sense.
Although the performance improvements against existing methods are not that significant, the stable improvements on multiple benchmarks demonstrate the effectiveness of the proposed method.

---

> ### Author Response · Authors · 2021-11-21
> **Response to Reviewer 3**
>
> We thank the reviewer for the thoughtful feedback. We are encouraged that you found our approach interesting and elegant showing performance gains on multi-task benchmarks. We address the reviewer's concern below:
>
> **The performance on some benchmarks is not that good, such as the results on CELEBA**: We agree that for some benchmarks the gap between the proposed approach and state-of-the-art (SOTA) methods is not significant. However, our approach demonstrates consistent improvement over SOTA baseline methods on the majority of the multi-task learning problems such as multi-digit and multi-label classification, scene understanding, and visual localization. The results might be improved further by finding a better set of hyperparameters.

---

### Official Review · Reviewer_nRWJ · 2021-11-02

**Correctness:** 2
**Technical Novelty And Significance:** 3
**Empirical Novelty And Significance:** 2
**Recommendation:** 5
**Confidence:** 5

**Main Review:**

The main strength of the paper is the introduction of the dominance rate, which allows the approach to avoid task overfitting. Another strength of the paper is the  technical presentation that is quite detailed. The z-alignment is also a strong point of the paper due to the solution to the limitation of the linear scaling of the theta-alignment counterpart.

Regarding the weaknesses of the paper I would point out the first claim that the method provides consistent improvements over all tasks. This claim is not backed up by the experiments. First at Table 2 the authors claim a noticeable improvement of 0.06% for Task-L and 0.16% for Task-R. I believe the results difference to the SOTA method are marginal and, consequently, hard to draw any conclusions. Besides that I suggest the authors to include the work from Lee et al. [1] which proposes a single gradient step update for task balancing that shows better results than pareto-MTL and GradNorm with much higher differences than 0.06 and 0.16%, respectively.

[1] Sungjae Lee and Youngdoo Son. Multitask learning with single gradient step update for task balancing. arXiv preprint arXiv:2005.09910, 2020.

The Camera relocalization results shows a slight improvement over the compared methods for orientation. Nervertheless, I believe the 15% orientation improvement over GradNorm is not enough due to the 25% performance degradation when compared to GradNorm. Besides that I missed the reference to Radwan et al. [2] which shows much more accurate results for the 7-SCENES dataset. The compared and presented results are too far from been competitive with the cited paper.

[2] Noha Radwan, Abhinav Valada, Wolfram Burgard. VLocNet++: Deep Multitask Learning For Semantic Visual Localization And Odometry
IEEE Robotics And Automation Letters (RA-L), 3(4):4407-4414, 2018.

With the inclusion of the referred papers will be possible to better verify the claims related to performance and the comparison to non MOO methods make the method a more generalist approach.

Another suggestion is related to benchmark the proposed method in a high cardinality dataset (hundreds of tasks). Datasets like "PubChem BioAssay Dataset Study" count with 128 tasks that is ideal to check the scale limitations of theta-alignment and the z-alignment capabilities.


**Summary Of The Paper:**

The authors aim to perform gradient task balancing on MTL setting by aligning independent components of the training objective. They claim to be able to handle large differences in gradient magnitude. Not choosing the faster learning rate dominating direction prevent task overfitting. The main contribution is based on the analysis of the individual components of the gradients instead of the individual gradients using the dominance rate. Tasks should have small dominance coefficients. Through a series of experiments the authors show the proposed technique is competitive with other multi-objective optimization approaches.

**Summary Of The Review:**

The paper technical contribution is interesting, however the experimental section show the performance gains can be marginal and claims regarding results should be revised. Additional references and experiments are needed to draw conclusions regarding the actual empirical contributions of the method.

---

> ### Author Response · Authors · 2021-11-21
> **Response to Reviewer 2**
>
> Thank you very much for your detailed feedback. We are encouraged that you found our work technically sound, interesting, and detailed. We address the concerns in the order mentioned in the review.
>
> **I believe the results difference to the SOTA method are marginal and, consequently, hard to draw any conclusions**. It is true that it is quite challenging to outperform the state-of-the-art MTL approaches by a significant margin. However, if you compare our method with the counterparts with similar computational costs/complexity, e.g. Z-aligned vs. Uncertainty or $\theta$-aligned vs. GradNorm [7], the proposed methods perform on par or better on the majority of MTL benchmarks.
>
> **I suggest the authors to include the work from Lee et al. [1] that shows better results than pareto-MTL**. We added the reference to this work [1] to our updated version of the manuscript. The paper looks interesting, however, we do not fully agree with your point regarding the higher difference in performance on Multi-MNIST dataset. The absolute numbers presented in Table 1 (p.19) of the paper are questionable since we can hardly believe that state-of-the-art MTL methods such as [6, 7] can achieve only about 90\% accuracy on both tasks if properly evaluated/tuned. As it is shown in [4] and our paper the accuracy for these methods are about 96--97\% and 95\% on Task-L and Task-R respectively.
>
> **The Camera relocalization results shows a slight improvement... Besides that I missed the reference to Radwan et al. [2] which shows much more accurate results**. We thank the reviewer for this suggestion, we added this reference to our manuscript. Indeed, VlocNet++ [2] outperforms the original PoseNet [3] model presented in 2015. However, the main idea here was to compare different gradient-based multi-task learning techniques using a unified model for camera pose regression. We would like to emphasize that the proposed approach is architecture-agnostic and can be seamlessly integrated into any image-based localization models. We chose the PoseNet model since it is a light-weight model and its implementation is publicly available (in contrast to [2]). Moreover, we also report the growth rate, $\mathcal{R}$ as a measure of improvement of multi-task learning methods over the baseline (i.e. a uniformly weighted sum of individual loss objectives). It is defined as $\mathcal{R} = 1 - \frac{P_i}{P}$ where $P_i$ is the mean translation (orientation) error of a MTL method; $P$ is the mean translation (orientation) error of the baseline. Table 4 shows that the proposed MTL approach is able to improve both orientation and translation localization performance over the baseline in contrast to other strong MTL counterparts.
>
> **Another suggestion is related to benchmark the proposed method in a high cardinality dataset (hundreds of tasks). Datasets like "PubChem BioAssay Dataset Study" count with 128 tasks that is ideal to check...** Theoretical complexity of the proposed $\theta$-alignment and Z-alignment approaches as well as the baseline MTL methods is provided in Table 1. We evaluate the proposed approach on all of the traditional benchmarks which are widely used in multi-task learning [4,5]. Moreover, we propose a new MTL benchmark, camera relocalization, and provide a thorough evaluation of SOTA MTL methods and our approach on it. We verify our methods on a high cardinality multi-label classification task, CelebA having 40 different attributes [4,5]. We agree that considering a benchmark with an even larger number of tasks (e.g. PubChem BioAssay Dataset) might be an interesting research direction. However, due to the limited time budget and a limited number of pages, it is not feasible to conduct thorough experiments and include new results in the paper during the rebuttal.
>
>
>
> 1. Lee et.al: Multitask learning with single gradient step update for task balancing. ArXiv:2005.09910, 2020
> 2. Radwan et.al: VLocNet++: Deep Multitask Learning For Semantic Visual Localization And Odometry IEEE Robotics And Automation Letters (RA-L), 2018
> 3. Kendall et.al: PoseNet: A Convolutional Network for Real-Time 6-DoF Camera Relocalization. ICCV 2015.
> 4. Sener et.al: Multi-Task Learning as Multi-Objective Optimization. NeurIPS 2018.
> 5. Yu et.al: Gradient Surgery for Multi-Task Learning. NeurIPS 2020.
> 6. Kendall et.al: Multi-Task Learning Using Uncertainty to Weigh Losses for Scene Geometry and Semantics. CVPR 2018
> 7. Chen et.al: GradNorm: Gradient Normalization for Adaptive Loss Balancing in Deep Multitask Networks. ICML 2018

---

> > ### Comment · Reviewer_nRWJ · 2021-12-02
> > **Thank you for your response**
> >
> > I want to thank the authors for the response. Regarding my suggestions I would like to point out that for the experiments, specially MultiMNIST I still think to say you surpass the baselines by a noticeable margin is too strong of a statement. Other datasets like cityscapes and CELEBA show the method can be competitive but not state-of-the-art. Regarding high cardinality dataset I agree the time is short for these experiments.  Overall I think the technical contribution is interesting, however the experimental section show performance gains that can be considered marginal.

---

> > > ### Author Response · Authors · 2021-12-03
> > > **Thank you for your suggestion**
> > >
> > > Thank you for going through our response and getting back to us. Regarding your remaining concern: We agree with you that the statement of surpassing baselines on page 7 in the MultiMNIST paragraph should be tuned down. In its current form, it also does not keep the focus on the relevant message of the experiments overall, which is robust improvement across different MTL problems. We will revise the message in the introductory paragraph of Sec. 5 and reflect these changes in Sec. 5.2 as you suggested.

---

### Official Review · Reviewer_PcEb · 2021-11-03

**Correctness:** 3
**Technical Novelty And Significance:** 3
**Empirical Novelty And Significance:** 2
**Recommendation:** 8
**Confidence:** 3

**Main Review:**

### Strengths

* The authors deal with a very interesting problem, that of multi task learning in the presence of objectives with a large difference in gradient magnitudes
* They propose a way of quantifying the imbalance "The rate of dominance" and use it to propose an elegant solution based on the SVD of the gradients of each loss, and an approximate version that is only computed on the shared parameters.
* the authors compare against recent works and show some empirical gains on sommon multi-task scenarios

### Weaknesses, questions and notes

1) In the illustration of figure 1, we see that there is a common solution for the two tasks in parameter space (towards the top left corner). Is it always possible to "guarantee consistent improvement over all tasks"? what if two tasks are very different (eg learning jointly for rotation invariance and for 6-9 digit recognition).

2) In section 3 the authors say: " task-specific parameters are independent of each other and shared parameters, and therefore have a single gradient, and thus can be omitted without loss of generality." - Isn't this assumption very strong? I understand (and it seems) that that such an approximation indeed works, yet, the fact is that this might not be true in practice

3) Notation in Fig 2 is defined way after the figure is referenced in the intro of sec 4 and this makes the fig hard to understand. Adding some explanation for $u_i$ and $\sigma_i$ would make it better.

4) it is unclear to me what "vector w denotes a pre-selected task preference" means. Is this a hyperparameter weight per task? how is it set?

5) Beyond differences in the applications and tasks, the following is a missing related work:
> Wang, Weiyao, Du Tran, and Matt Feiszli. "What makes training multi-modal classification networks hard?." Proceedings of the IEEE/CVF Conference on Computer Vision and Pattern Recognition. 2020.
Authors should discuss the relation of the proposed dominance to the overfitting-to-generalization-ratio (OGR) from Wang et al.
Ideally, a comparisson to GradientBlending would be great.

6) In algorithm 1 line 13: why are only the $\theta_{sh}$ parameters updated? how about the update of $\theta_i$

7) the task specific parameters seem to come as "heads" on top of a common shared backbone g() - Would this approach extend to works that use adaptors thoughout the net as in
> Housby et al., Parameter-Efficient Transfer Learning for NLP
> Pfeiffer et al., AdapterFusion: Non-Destructive Task Composition for Transfer Learning

8) Shouldnt the subscript  G_theta in definitin 3 be G_Z ?

9) I did not fully validate the correctness of the proofs of the theorems in the appendix. I think that a 1-2 sentence discussion is however needed on the theorems in the text, eg for Theorem 1, the second condition contains a lot of ndefined notation and it is really hard to understand. This relates with point 4 above and could clarify the role of vector w

10) Why is  Z-aligned, an efficiency-motivated approximation (In conclusions the authors claim that this was created "to ensure practicality"), better than $\theta$-aligned in many (most) cases?

11) Another work that can be cited and discussed is
> Lu, Jiasen, et al. "12-in-1: Multi-task vision and language representation learning." Proceedings of the IEEE/CVF Conference on Computer Vision and Pattern Recognition. 2020.


**Summary Of The Paper:**

The paper presents an approach for better multi task learning in the presence of objectives with a large difference in gradient magnitudes. They study the problem and define a "The rate of dominance" metric that is based on an eigenvalue ratio of the gradient tensors. they then use this dominance metric as a coefficient to better "balance" gradients across tasks and propose an elegant solution based on the  an approximate version that is only computed on the shared parameters. They report results on common multi-task settings, and either perform on par or outperform other recent methods.

==== After rebuttal ====

I want to thank the authors for a complete and detailed response. The answers make sense to me, and I am still in favour of acceptance - I raise my score to Accept.



**Summary Of The Review:**

Overall an interesting and elegant approach, that shows some gains. There are weaknesses, discussed above; looking forward to the rebuttal from the authors.

---

> ### Author Response · Authors · 2021-11-21
> **Response to Reviewer 1 (1 of 2)**
>
> We thank the reviewer for the thoughtful feedback as well as the suggestions for improvement. We are encouraged that you found our approach interesting and elegant showing performance gains on multi-task benchmarks. We address your concerns below:
>
> **In the illustration of figure 1, we see that there is a common solution for the two tasks... Is it always possible to "guarantee consistent improvement over all tasks"? what if two tasks are very different...** Since in terms of the original problems it is impossible to guarantee a consistent improvement of all problems simultaneously without using the Pareto conditions, in our work we aim to find a set of consistent orthogonal problems linearly composed of the original loss functions at each step. This set is the closest in terms of the Frobenius norm to the set of initial problems (their gradients). Given a new set of orthogonal tasks, it is possible to guarantee the improvement of all tasks at the same time.
>
> **In section 3 the authors say: " task-specific parameters are independent of each other and shared parameters, and therefore have a single gradient, and thus can be omitted without loss of generality." - Isn't this assumption very strong?** Since first-order gradient-based optimization approaches are linear, the interaction outside of linearity can not be accounted for. In practice, gradients over task-specific parameters and shared parameters are pairwise linearly independent. This is why we used this approximation. We have updated the manuscript accordingly to clarify this.
>
> **Adding some explanation for $u_i$ and $\sigma_i$ to Figure 2 would make it better**. We thank the reviewer for this suggestion. We have updated our manuscript by adding notation to the caption of Figure 2.
>
> **it is unclear to me what "vector w denotes a pre-selected task preference" means. Is this a hyperparameter weight per task? how is it set?**. Correct, the parameter $\mathbf{w}$ is a hyperparameter which defines the importance of each task. In our multi-task learning setup, since all the tasks are equally important, $\mathbf{w}$ is initialized to $\mathbf{1}$ and kept unchanged during training. However, if we focus on one particular task, the task weights are different which leads to auxiliary task learning. Extending the proposed approach to auxiliary task learning is a very interesting future research direction. We thank the reviewer for this suggestion.
>
> **Beyond differences in the applications and tasks, the following is a missing related work [1]. Authors should discuss the relation of the proposed dominance to the overfitting-to-generalization-ratio (OGR) from Wang et al**. We thank the reviewer for pointing this out. Somehow, this work was under our radar. We have updated our manuscript accordingly (Section 4, after Definition 1). In that paper, the authors propose a metric, the overfitting-to-generalization ratio (OGR) to quantitatively measure the significance of overfitting in joint training of multi-modal networks. The metric is defined by using an assumption that target distribution (a test data set) is well approximated by a held-out validation data set. Technically speaking, in order to compute OGR, they require to have the ground-truth gradient $\nabla \mathcal{L}^*$ (Equation 9 in Supplementary material of the original paper) which is not available in practice. In contrast, our measure does not require such strong prior and solely based on gradients with respect to the training set. Moreover, our approach is applied to not only classification but also regression.
>
> **In algorithm 1 line 13: why are only the parameters $\theta_{sh}$ updated? how about the update of $\theta_i$**. Right! Both parameters $\theta_{sh}$ and $\theta_i$ should be updated during the model training. The update rule for $\theta_i$ is presented in line 2 of the original Algorithm.
>
> **... the task specific parameters seem to come as "heads" on top of a common shared backbone g() - Would this approach extend to works that use adaptors thoughout the net as in [2] and [3]**. Yes, you are absolutely right. Both works [2, 3] propose different neural network architectures to efficiently combine information from several tasks. Our $\theta$-alignment approach is invariant to the model architecture and therefore can be applied to any setup in which more than one task is present. Z-aligned only applies to encoder-decoder architectures. Therefore, the proposed methods can be seamlessly integrated into Adaptors. Moreover, our methods are invariant to the number of problems, which makes it possible to add new tasks on the fly without changing the algorithm.
>
> **Shouldn't the subscript $G_{\theta}$ in definition 3 be $G_Z$**? Thank you very much! This is definitely a typo that is fixed in the updated version of the manuscript.

---

> > ### Author Response · Authors · 2021-11-21
> > **Response to Reviewer 1 (2 of 2)**
> >
> > **... I think that a 1-2 sentence discussion is however needed on the theorems in the text**. Although we provide the notation of mathematical symbols used in the theorems in the appendix, we agree that some clarifications in the main part are needed. We slightly changed Theorem 1 and Theorem 2 in the updated version of the paper to make them clearer.  Thank you very much!
> >
> > **Why is Z-aligned, an efficiency-motivated approximation (In conclusions the authors claim that this was created "to ensure practicality"), better than $\theta$-aligned in many (most) cases**. We associate this fact with a more stable solution of the Procrustes problem in the case of a Z-aligned gradient due to the smaller size of the tensors.
> >
> > **Another work that can be cited and discussed is [4]**. We thank the reviewer for this interesting reference. Lu et al. [4] propose a heuristic method to solve the problem of a task domination in multi-task learning. Although showing excellent results, the approach requires large computational costs to find an optimal set of hyperparameters to balance the training procedure. In contrast to [4], our method can automatically balance training in multi-task systems and significantly improve computational burden. However, combining [4] and our approach is an interesting research question for feature work.
> >
> > 1. Wang et al: What makes training multi-modal classification networks hard? CVPR 2020
> > 2. Housby et al: Parameter-efficient transfer learning for NLP. ICML 2019
> > 3. Pfeiffer et al: AdapterFusion: Non-destructive task composition for transfer learning. ECACL 2021.
> > 4. Lu et al: 12-in-1: Multi-task vision and language representation learning. CVPR 2020.

---

> > > ### Comment · Reviewer_PcEb · 2021-11-30
> > > **Thank you for a detailed response**
> > >
> > > I want to thank the authors for a complete and detailed response. The answers make sense to me, and I am still in favour of acceptance.
> > >
> > > My only remaining comment is that I would suggest the authors to discuss more in the text the performance of the Z-aligned variant. The comment/response above that it offers "a more stable solution of the Procrustes problem [...] due to the smaller size of the tensors." should be incorporated and maybe even further discussed. I would also remove the "To ensure practicality" argument for the same variant, as, sure, maybe it was computation the reason that this was created but in the end it outperforms  $\theta$-aligned-- it would be nice to discuss possible reasons more extensively.

---

> > > > ### Author Response · Authors · 2021-11-30
> > > > **Thank you for the suggestion!**
> > > >
> > > > Thank you for the great suggestion and for reading our response. Yes, our assumption on why the Z-aligned approach works better than the $\theta$-aligned is based on Weyl's theorem [1, 2]. The theorem provides a bound on the largest distance between exact and perturbed eigenvalues of a Hermitian matrix computed by any backward stable algorithm, such as QR. Therefore, the error in eigenvalue estimation is defined as $|\sigma_i - \hat\sigma_i| \leq \| E \|_2$, where $E$ is a symmetric error matrix. The task dimensionality is defined by the size of matrix $E$. The $\theta$-aligned method has a larger task dimensionality compared to the Z-aligned, which leads to a larger error in SVD and thus a less accurate solution of the Procrustes problem. We will add this clarification to the revised version of our manuscript.
> > > >
> > > > 1. Nakatsukasa Y. (2010). Absolute and relative Weyl theorems for generalized eigenvalue problems. Linear Algebra and its Applications, 432:  242–-248.
> > > > 2. Demmel J. (1997). Applied Numerical Linear Algebra, SIAM.

---

### Author Response · Authors · 2021-11-22
**Summary of modifications**

We thank the three anonymous reviewers for their feedback on our manuscript. We summarize the reviews as positive overall: The proposed methods were well received by all reviewers, and all reviewers pointed out that the approach was interesting and sensible.

We recapitulate the main idea of the paper, which hinges on the realization that improving upon the overall performance in multi-task learning can be formulated by conservatism in the individual learning steps. This gives rise to our elegant heuristic in terms of a *dominance rate* which allows the approach to avoid task overfitting. Our method is *general and applicable across MTL problems*, which also means that we do not aim for state-of-the-art performance in individual benchmark problems, for which specialized approaches have been proposed. Instead, we aim for robust improvement across different MTL problems and showcase this in the experiments (e.g., see updated Table 4, where we are the only method to improve upon both translation and orientation). Following the reviewer feedback, we have made this point clearer.

We have used the opportunity to revise our manuscript to reflect the reviewer feedback and comments. The main changes are related to clarifications requested by reviewer PcEb and modifications to the experiments to better highlight the properties of the proposed approach (as brought up by reviewers nRWJ and RbSw). We have separately addressed the concerns of each reviewer below.

---

### Decision · Program_Chairs · 2022-01-20

**Decision:**

Reject

**Comment:**

This paper presents work on multi-task learning.  The reviewers appreciated the method based on SVD of loss gradients.  However, concerns were raised regarding empirical effectiveness and overall impact.  The reviewers considered the authors' response in their subsequent discussions.  While the methods are interesting, the concerns over their effectiveness would need to be more thoroughly addressed in order to improve the impact of the paper.  As such, it is encouraged that the authors take these suggestions into account in preparing a new version of the paper for a future submission.